# The heat shock protein LarA activates the Lon protease in response to proteotoxic stress

Deike J. Omnus [1,4], Matthias J. Fink [1,4], Aswathy Kallazhi [1],
Maria Xandri Zaragoza[1], Axel Leppert [2], Michael Landreh [2,3] &
Kristina Jonas [1]✉

The Lon protease is a highly conserved protein degradation machine that has critical regulatory and protein quality control functions in cells from the three domains of life. Here, we report the discovery of a α-proteobacterial heat shock protein, LarA, that functions as a dedicated Lon regulator. We show that LarA accumulates at the onset of proteotoxic stress and allosterically activates Lon-catalysed degradation of a large group of substrates through a five amino acid sequence at its C-terminus. Further, we find that high levels of LarA cause growth inhibition in a Lon-dependent manner and that Lon-mediated degradation of LarA itself ensures low LarA levels in the absence of stress. We suggest that the temporal LarA-dependent activation of Lon helps to meet an increased proteolysis demand in response to protein unfolding stress. Our study defines a regulatory interaction of a conserved protease with a heat shock protein, serving as a paradigm of how protease activity can be tuned under changing environmental conditions.

All cells must be able to adjust their proteome composition in response to internal and external signals. In particular free-living microorganisms that inhabit strongly fluctuating environments dynamically remodel the proteome to ensure adaptation[1]. Intracellular proteolysis plays a critical role in this process as it regulates the amounts of specific functional proteins and removes un- and misfolded proteins that accumulate under various stress conditions[2,3].

In prokaryotes and in the mitochondria, chloroplasts and peroxisomes of eukaryotic cells, proteolysis is carried out by ATP-dependent proteases of the Clp, Lon, Hsl and FtsH families[2,4]. These macromolecular machines consist of an AAA+ (ATPases Associated with diverse cellular Activities) unfoldase module that unfolds protein substrates and subsequently translocates them into a proteolytic chamber formed by a peptidase module. Protein degradation must be highly specific and is therefore tightly regulated. Most proteases recognise their substrates by short sequence tags, so-called degrons,

that often occur at the termini of protein substrates and establish the initial contacts between protease and substrate[4]. Additionally, efficient substrate recognition and degradation often involves accessory regulatory proteins, including adaptors that bridge the interaction between protease and specific substrates as well as allosteric regulators that modulate protease activity by binding to a regulatory site that is distinct from the affected substrate binding site[4,5].

Several protease adaptors have been identified in different species, most of which act on proteases of the Clp protease family. A prominent example is the SspB adaptor protein, which mediates ClpXP-dependent degradation of SsrA-tagged polypeptides generated by trans-translation[6]. Other examples include the ClpXP-adaptors CpdR, RcdA and PopA in the α-proteobacterium *Caulobacter crescentus*[7], the ClpCP adaptor MecA and the ClpXP adaptor YjbH in *Bacillus subtilis*[8,9] as well as ClpS, an adaptor of the ClpAP protease in most Gram-negative bacteria[10].

[1]Science for Life Laboratory and Department of Molecular Biosciences, The Wenner-Gren Institute, Stockholm University, Svante Arrhenius väg 20C, Stockholm 10691, Sweden. [2]Department of Microbiology, Tumor and Cell Biology, Karolinska Institutet, Solnavägen 9, 17165 Solna, Sweden. [3]Department of Cell and Molecular Biology, Uppsala University, Box 596, 751 24 Uppsala, Sweden. [4]These authors contributed equally: Deike J. Omnus, Matthias J. Fink. ✉e-mail: kristina.jonas@su.se

For the protease Lon, the number of identified adaptors and dedicated regulators is surprisingly small so far, even though Lon was the first ATP-dependent protease to be discovered and is widely conserved in the three domains of life[11]. Dysregulation or malfunction of mammalian Lon has been associated with ageing, neurodegenerative diseases and cancer[12]. In bacteria, Lon affects a variety of important cellular functions including cell cycle progression, stress responses, cell differentiation, pathogenicity and antibiotic tolerance[13]. Lon is also a major protease responsible for the degradation of un- and misfolded proteins[11,14–16] and it was shown to recognise misfolded proteins by its affinity to sequences rich in hydrophobic and aromatic residues that normally are buried within folded proteins[17]. Early work already demonstrated allosteric regulation of Lon. In vitro experiments showed that certain Lon substrates, more specifically the degron tags that are exposed on those substrates, can allosterically activate Lon proteolysis[18,19]. In *C. crescentus*, allosteric activation of Lon by unfolded protein substrates was shown to enhance clearance of the DNA replication initiator DnaA at the onset of proteotoxic stress[20].

Despite its important regulatory functions, so far only two specific proteins that modulate Lon-mediated proteolysis have been described. The protein SmiA in *B. subtilis* mediates Lon-dependent degradation of the master regulator of flagella biosynthesis, SwrA, via a direct interaction with the C-terminus of SwrA, and thus shows properties of a classical adaptor[21–23]. In *Yersinia pestis* and *Salmonella enterica* serovar Typhimurium, the protein HspQ functions as a specificity-enhancing factor of Lon[24,25], which does not act by a substrate delivery mechanism, but instead allosterically activates Lon proteolysis of certain substrates[24].

Here, we report the discovery and characterization of LarA, a dedicated and stress-induced Lon regulator in *C. crescentus* that activates Lon proteolysis of a broad range of substrates. We find that LarA temporarily accumulates in response to temperature upshift and strongly activates Lon substrate degradation via specific amino acids within a C-terminal degron sequence that also contributes to LarA's own degradation by Lon. We demonstrate that high LarA levels result in toxic overactivation of Lon, thus underscoring the need for timely restricted presence of LarA and precisely controlled Lon activity.

## Results

### A trapping approach identifies LarA (CCNA_03707) as a putative Lon substrate

To identify substrates and potentially regulatory proteins of Lon in *C. crescentus*, we employed a protease trap approach[26] using strains lacking a functional copy of native *lon* (Δ*lon*) that expressed either wild type Lon (Lon^WT) or an active-site mutant of Lon (Lon^TRAP), both harbouring a C-terminal Twin-Strep-tag for purification (Fig. 1a). In the Lon^TRAP variant, the conserved catalytic serine residue S674 required for peptidase activity is replaced by an alanine residue (Fig. 1a, Supplementary Fig. 1a), generating a Lon variant that is capable of binding, unfolding and translocating substrates without degrading them[27], thus making it a tool for the identification of protease-bound proteins[24,28]. After verifying that Lon^TRAP was proteolytically inactive by monitoring the levels of the known Lon substrate DnaA (Supplementary Fig. 1b), we purified in parallel Lon^WT and Lon^TRAP via their Twin-Strep-tags and used cell lysates from Δ*lon* cells as control (no Lon). DnaA was clearly enriched in the elution fractions of the Lon^TRAP purification (Supplementary Fig. 1c), but not with Lon^WT or the control sample not expressing any *lon*. Subjecting elution fractions to tandem mass tag (TMT) labelling followed by mass spectrometry analysis, led to the identification of proteins that specifically eluted with Lon^TRAP (Fig. 1b, Supplementary Dataset 1). Among the enriched proteins were several previously validated Lon substrates, including DnaA, StaR and FliK, as well as proteins that were in a previous study classified as putative Lon substrates (Fig. 1b)[20,29]. CCNA_03707 (hereafter referred to as LarA for Lon activity regulator A) (Fig. 1b), was one of the most enriched

proteins, which indicates a strong physical interaction with Lon and additionally suggests it is a Lon substrate. LarA is conserved in a large group of α-proteobacteria and is annotated as a DUF1150 family protein, but its function remained uncharacterized.

### LarA is a Lon substrate

To test if LarA is indeed a Lon substrate, we analysed the degradation of 3xFLAG-tagged LarA versions in vivo. Western blot analysis showed that the levels of N-terminally 3xFLAG-tagged LarA (F-LarA) gradually decreased after blocking protein synthesis by addition of the antibiotic chloramphenicol (Fig. 1c), demonstrating that F-LarA is degraded in wild type cells (WT). By contrast, in Δ*lon* cells F-LarA protein levels remained high (Fig. 1c), indicating that F-LarA degradation depends on Lon. C-terminally 3xFLAG-tagged LarA (LarA-F) showed slower degradation in wild type cells compared to F-LarA, but was again more stable in Δ*lon* cells (Fig. 1d). These results demonstrate that LarA is degraded in a Lon-dependent manner in vivo and suggest a role of the C-terminus of LarA in recognition by Lon.

To directly test whether LarA is degraded by Lon, we performed in vitro degradation assays using purified Lon and LarA. LarA was robustly degraded when incubated in the presence of Lon and ATP, but remained stable in reactions lacking ATP and its regeneration system (Fig. 1e, f), demonstrating that LarA is degraded by Lon in an ATP-dependent manner. Together, these data show that LarA is a bona fide Lon substrate.

### LarA inhibits growth in a Lon-dependent manner

To gain insight into the function of LarA, we analysed the consequences of LarA overexpression on cell growth. Xylose-inducible overexpression of F-LarA from a medium copy vector led to a clear growth defect compared to a control strain carrying the empty vector (Fig. 2a). Strikingly, the growth inhibitory effect of F-LarA overexpression was completely abolished in Δ*lon* cells (Fig. 2b), demonstrating that F-LarA only inhibits growth when Lon is present. The growth defect caused by F-LarA overexpression was also absent in Δ*lon* cells harbouring only proteolytically inactive Lon^TRAP (Fig. 2c), whereas Δ*lon* cells expressing *lon^WT* exhibited a notable growth defect in response to F-LarA overexpression (Fig. 2d). This demonstrates that the growth inhibitory effect of F-LarA overexpression depends on the proteolytic activity of Lon.

Based on these results, we hypothesised that LarA functions as an activator of Lon and that excess levels of LarA lead to an upregulation of Lon-dependent degradation that causes growth inhibition. If this hypothesis was true, elevating the levels of Lon in addition to LarA overexpression should lead to an even stronger growth inhibition than LarA overexpression alone. Consistent with this idea, xylose-inducible co-overexpression of *lon^WT* and *F-larA* in an otherwise wild type background, which still harbours functional *lon* at the native locus, arrested growth almost completely (Fig. 2e). The growth arrest was accompanied by an accumulation of cells with two fully replicated chromosomes indicative of a delay in cell division (Supplementary Fig. 2a) as well as a detectable reduction in total protein content, as early as 2 h after xylose addition (Supplementary Fig. 2b). Importantly, the severe growth phenotype was absent in a vector control strain overexpressing *lon* alone (Fig. 2e) and to a milder extent when co-overexpressing *F-larA* in combination with *lon^TRAP* (Fig. 2f). These data suggest that the levels of proteolytically active Lon are rate-limiting for the toxic effect induced by *F-larA* overexpression and strongly support our hypothesis that LarA acts as an activator of Lon-mediated degradation.

### LarA allosterically activates Lon-mediated degradation

To more directly assess if LarA is a Lon regulator, we monitored the effect of *F-larA* overexpression on the degradation kinetics of the known Lon substrate SciP[30], a G1-specific transcriptional regulator in *C. crescentus*[31]. Consistent with previous results, SciP was unstable and

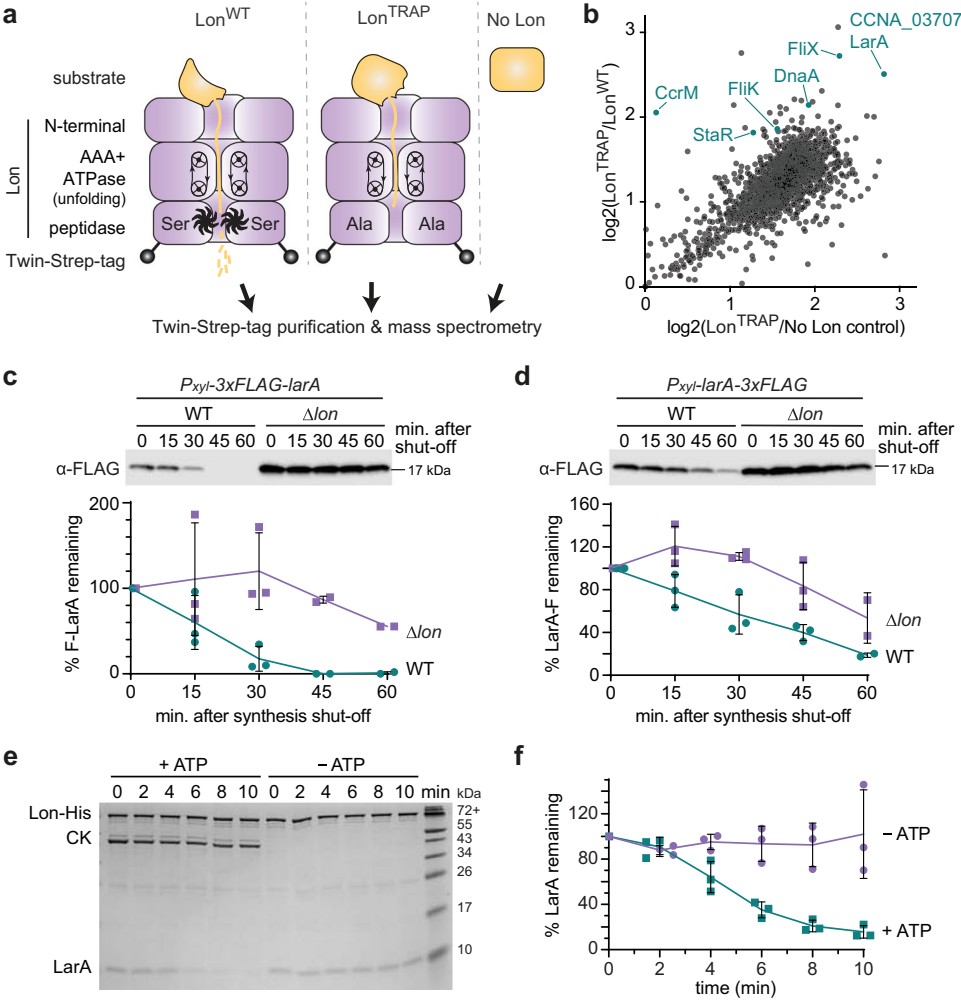

**Fig. 1 | A trapping approach identifies LarA as an interactor and substrate of Lon. a** Schematic representation of the constructs used for the trapping experiment. C-terminally Twin-Strep-tagged Lon^WT or Lon^TRAP were expressed from a xylose-inducible $P_{xyl}$ promoter in $\Delta lon$ cells. Extracts from $\Delta lon$ cells lacking Lon were used as control (No Lon). **b** Dot plot showing proteins enriched in the Lon^TRAP elution fraction compared to the Lon^WT and the No Lon elution fraction. Log2 values are based on averages of two independent experiments (Supplementary Dataset 1). Turquoise dots mark known/confirmed Lon substrates (FliK corresponds to CCNA_00944/45 in Supplementary Dataset 1) as well as CCNA_03707/LarA. **c** In vivo stability of N-terminally 3xFLAG-tagged LarA (F-LarA) in wild type (WT) and $\Delta lon$ cells. Quantifications show the mean values ± SD of F-LarA levels; $n = 3$ or 2 ($t = 45$, $t = 60$) biologically independent samples. **d** Same as in (**c**), but with C-terminally 3xFLAG-tagged LarA (LarA-F). LarA-F levels are presented as mean values ± SD; $n = 3$ or 2 ($t = 60$) biologically independent samples. **e** In vitro degradation of LarA by Lon. The reactions contained 3.0 μM LarA and 0.125 μM Lon-His hexamer with (+ATP) or without (−ATP) ATP regeneration system (ATP, creatine phosphate, creatine kinase [CK]). **f** Quantifications showing LarA protein levels (normalised to Lon-His levels) as mean values ± SD; $n = 3$ independent experiments. Source data are provided as a Source Data file.

showed a half-life of approximately 20 minutes in the vector control strain (Fig. 3a). In support of our hypothesis that LarA is a Lon activator, SciP degradation was notably increased in cells overproducing F-LarA, resulting in a half-life of less than 10 minutes (Fig. 3a), and a similar result was obtained when overexpressing untagged LarA (Supplementary Fig. 3a). The destabilisation of SciP in *larA*-overexpressing cells was also reflected in reduced SciP steady-state levels after 2 and 4 h of induction of *F-larA* expression (Supplementary Fig. 3b).

To ascertain that LarA directly regulates Lon activity, we determined the effects of purified LarA on Lon-mediated degradation of 6xHis-tagged SciP (His-SciP) in vitro. Consistent with our in vivo data, LarA strongly stimulated in vitro His-SciP degradation (Fig. 3b). In the absence of LarA, His-SciP was degraded with a half-life of approximately 8 minutes. However, when LarA was present, the half-life of His-SciP was shorter than 2 minutes (Fig. 3b). In agreement with our results that LarA is a substrate itself, LarA levels declined over time, however more slowly than His-SciP levels (Fig. 3b).

To gain deeper insights into the mechanism by which LarA activates Lon, we characterised the kinetics of LarA-activated His-SciP degradation by Lon. First, we determined Lon-mediated degradation rates of His-SciP over a range of LarA concentrations and found that His-SciP degradation concomitantly increased with rising LarA concentrations up to a concentration of 2 μM LarA, when the degradation rate was 5.5-fold enhanced compared to reactions lacking LarA (Fig. 3c, Supplementary Table 1). Further increasing the concentration of LarA led to only a subtle decrease in degradation rate. Importantly, even at 7.5-fold excess of LarA (15 μM) over His-SciP, degradation remained at a high rate, indicating that LarA does not notably compete with His-SciP for binding to Lon.

Having determined the concentration at which LarA maximally activates Lon-dependent degradation of His-SciP, we investigated Lon degradation rates as a function of His-SciP concentration in the absence or presence of LarA at this concentration. Without LarA, His-SciP degradation rates followed typical Michaelis-Menten kinetics with a $V_{max}$ of 7.6 min⁻¹ Lon₆⁻¹ at saturated substrate concentrations and a

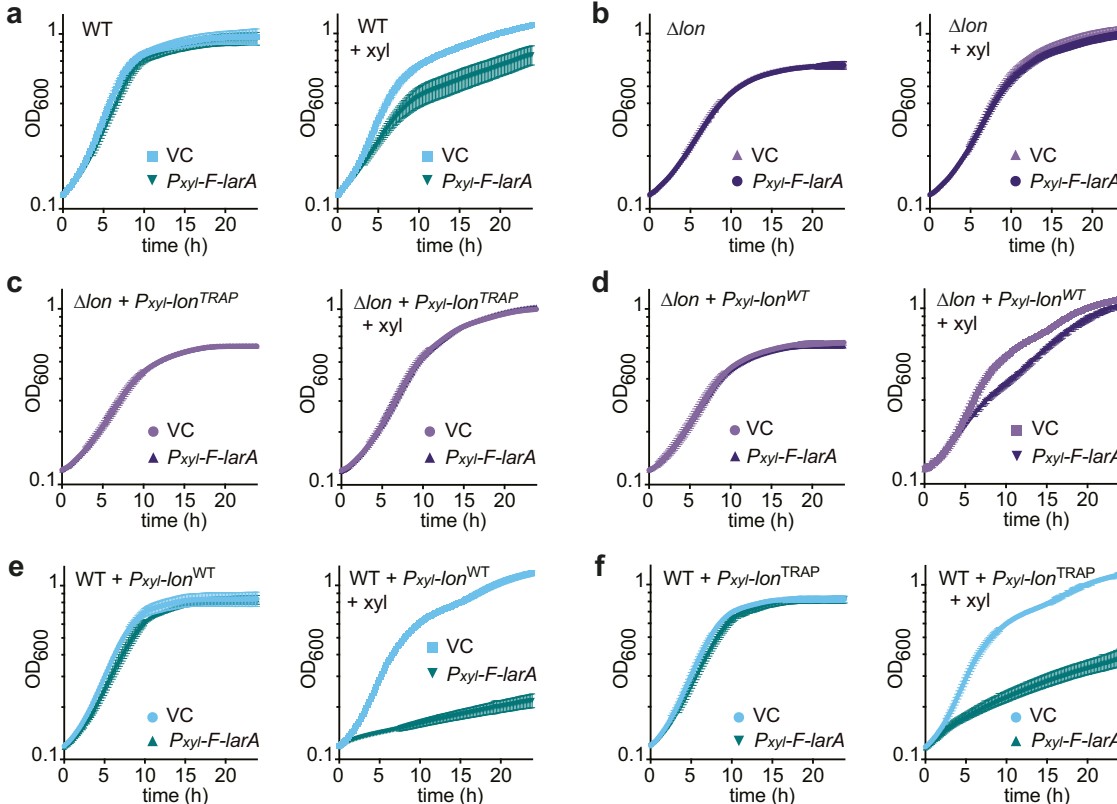

**Fig. 2 | LarA inhibits growth in a Lon-dependent manner. a** Growth curves of wild type *C. crescentus* (WT) harbouring either an empty vector (VC) or a plasmid carrying $P_{xyl}$-*3xFLAG-larA* ($P_{xyl}$-*F-larA*). Cultures were either grown under non-inducing conditions (left panel) or $P_{xyl}$-inducing conditions (+xyl; right panel). Growth curves display means ± SD; *n* = 9 biologically independent cultures from three independent experiments. **b** Same as (**a**) but in a *Δlon* background and *n* = 6 biologically independent cultures from two independent experiments. **c** *Δlon* cells with *lon*^TRAP-*Twin-Strep-tag* integrated on the chromosome under control of the $P_{xyl}$ promoter and harbouring either an empty vector (VC) or a plasmid carrying $P_{xyl}$-*3xFLAG-larA*. Cultures were grown under non-inducing (left panel) or inducing conditions (+xyl). **d** Same as in (**c**) but with *lon*^WT-*Twin-Strep-tag* integrated on the chromosome under control of the $P_{xyl}$ promoter. **e, f** Same as (**c, d**) but in a WT background, instead of *Δlon*. Growth curves in (**c**–**f**) display means ± SD; *n* = 3 biologically independent cultures from three independent experiments. Source data are provided as a Source Data file.

$K_m$ of 9.3 µM (Fig. 3d, Supplementary Table 2). The presence of LarA significantly increased the $V_{max}$ to 15 min$^{-1}$ Lon$_6$$^{-1}$, reduced the $K_m$ to 1.8 µM and resulted in a sigmoidal curve with a Hill coefficient of 1.9 (Fig. 3d, Supplementary Table 2), indicative of positive cooperativity. Based on these parameters, we calculated the reaction rate and the catalytic efficiency of Lon during His-SciP degradation and found that upon LarA addition the reaction rate $k_{cat}$ was 2-fold and the catalytic efficiency 10-fold enhanced (Supplementary Table 2). These data strongly suggest that LarA does not merely enhance the affinity of Lon to SciP, as would be expected for a classical adaptor, but also stimulates Lon catalysis by binding to a distinct regulatory site that mediates allosteric activation of Lon.

Prompted by the strong effect of LarA on the proteolytic activity of Lon, we aimed to understand if LarA affects the unfoldase, *i.e.*, the ATPase activity of Lon. Incubation of Lon with either LarA or His-SciP by themselves, resulted in a comparable increase in ATPase activity (Fig. 3e). Importantly, the presence of LarA and His-SciP together, at equimolar concentrations, had a synergistic effect on Lon's ATPase activity resulting in more than 2-fold higher ATPase activity compared to the reactions containing just one of the proteins at the same total substrate concentration (Fig. 3e). This significant increase in ATPase activity was not seen when co-incubating Lon and His-SciP with LarA$^{Δ5}$ (Fig. 3e), a C-terminally truncated LarA variant that is deficient in activating Lon proteolysis (see section below for more details about this mutant). These data demonstrate that the LarA-dependent stimulation of Lon-dependent degradation correlates with an increased ATPase activity.

To further investigate the interaction between Lon and LarA we used native mass spectrometry (native MS), which can provide insights into stoichiometries of protein complexes[32]. To prevent LarA degradation during analysis, we utilized a Walker-B mutant of Lon (Lon$^{EQ}$), which is deficient in ATP hydrolysis and thus cannot unfold and translocate substrates to degrade them (Supplementary Fig. 4a)[33]. Subjecting Lon$^{EQ}$ itself to native MS analysis revealed a calculated molecular weight of 540,843 ± 530 Da agreeing with the predicted molecular weight of a Lon hexamer of 529,458 Da (Fig. 3f). Incubation of Lon$^{EQ}$ with equimolar concentrations of LarA resulted in an increase of the measured molecular weight by 57,216 Da, corresponding to six LarA molecules (predicted MW of 6 LarA: 56 076 Da) bound to one Lon$^{EQ}$ hexamer (Fig. 3f, Supplementary Fig. 4b). This mass shift did not change when LarA was supplied in 3-fold excess (Supplementary Fig. 4c). These data suggest that six LarA molecules stably bind to one hexameric Lon complex. Importantly, the mass shift was not seen when adding the C-terminally truncated LarA$^{Δ5}$ variant (Fig. 3f, Supplementary Fig. 4b). Moreover, our native MS data did not indicate complex formation between LarA and the substrate SciP under the tested conditions (Supplementary Fig. 4d).

Together, our data show that LarA binds to a distinct regulatory site on Lon which increases the catalysis rates of Lon as well as Lon's binding affinity to another substrate. This suggests that LarA acts as an allosteric regulator of Lon activity, rather than a tethering adaptor that bridges the interaction between Lon and substrates.

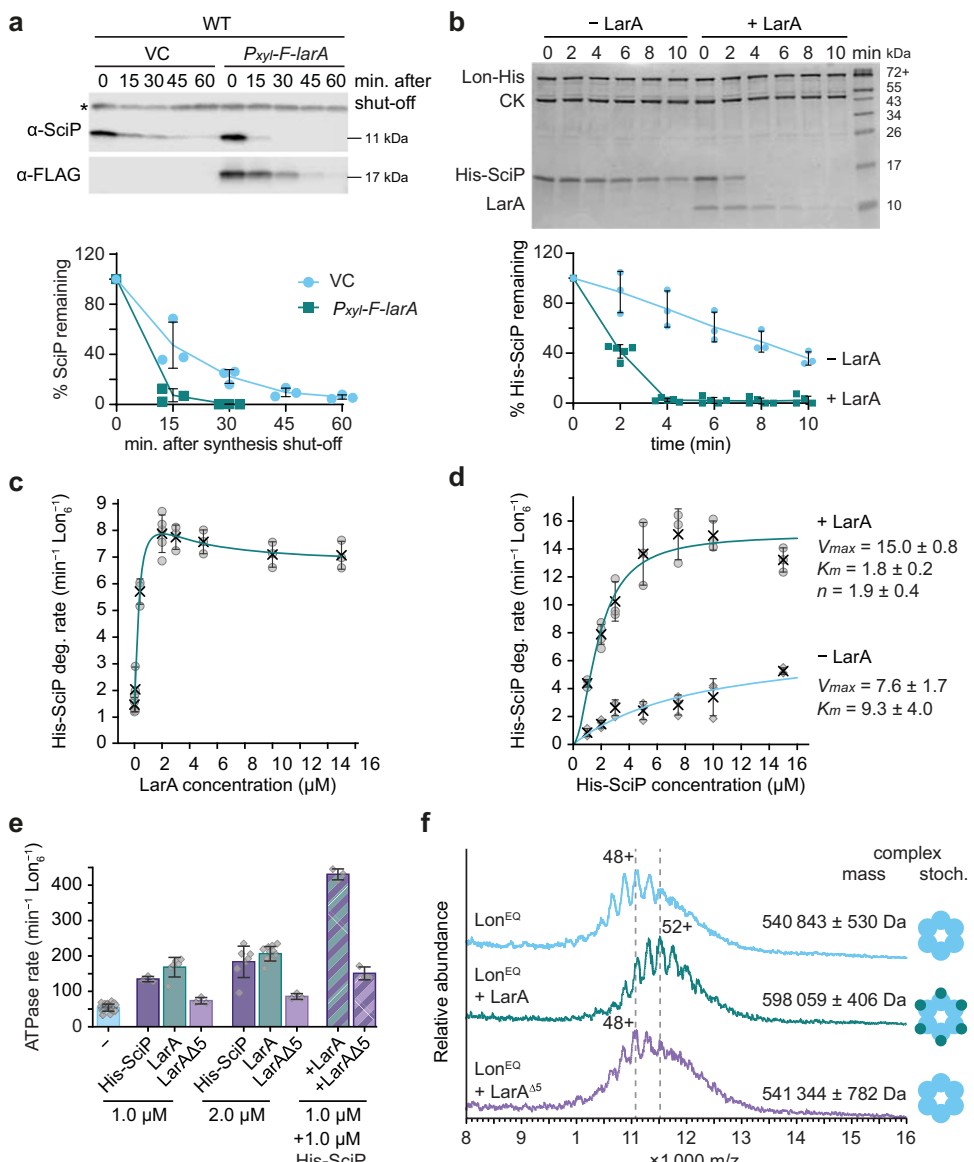

**Fig. 3 | LarA allosterically activates Lon-mediated degradation. a** In vivo stability of SciP and 3xFLAG-LarA (F-LarA) in WT cells carrying an empty vector (VC) or a plasmid carrying $P_{xyl}$-3xFLAG-larA ($P_{xyl}$-F-larA). Quantifications show the mean values ± SD of SciP levels; $n = 3$ biologically independent samples. **b** In vitro degradation of His-SciP (4 μM) by Lon-His hexamer (0.125 μM) in the presence (+LarA, 3 μM) or absence of LarA (−LarA). Creatine kinase [CK] was added for ATP regeneration. His-SciP protein levels (normalised to Lon-His levels) are presented as mean values ± SD; $n = 3$ (−LarA), 5 (+LarA) independent experiments. **c** Degradation rates of 2 μM His-SciP at increasing concentrations of LarA. All reactions contained 0.05 μM Lon-His hexamer. The curve represents a fitted equation considering activation and inhibition by LarA (see Methods). Values represent means ± SD; $n = 3$, 4 (0 μM LarA) or 5 (2 μM LarA) independent experiments. **d** Degradation rates at increasing His-SciP concentrations in the absence (−LarA) or presence of 2 μM LarA

(+LarA). Lon-His hexamer 0.05 μM was used. Curves were fitted to Michaelis-Menten and Hill equations (see Methods). Values represent means ± SD; $n = 3$, 4 (2 μM − LarA) or 5 (2 μM + LarA) independent experiments. **e** ATPase rates of Lon-His hexamer (0.05 μM) in the presence of SciP, LarA, LarA$^{\Delta5}$, either when added individually or in combination at the indicated concentrations. A reaction without substrate (−) is shown for comparison. Values represent means ± SD; $n = 3$, 6 (1 μM LarA), 7 (2 μM His-SciP), 12 (2 μM LarA) or 36 (−) independent reactions. **f** Native mass spectra of Lon$^{EQ}$ alone and of Lon$^{EQ}$ with either LarA or LarA$^{\Delta5}$. The main charge state as well as the experimentally determined molecular weight of the detected complex is indicated. Representative data are shown; $n = 3$ independent experiments. Full spectra of LonEQ + LarA and LonEQ + LarA$^{\Delta5}$ showing the charge states of unbound LarA and LarA$^{\Delta5}$ are shown in Supplementary Fig. 4. Source data are provided as a Source Data file.

## LarA interacts with Lon via a C-terminal degron that is critical for Lon activation

To identify sequence features in LarA that mediate the interaction with Lon, we turned our attention to the C-terminus of LarA. Our result that C-terminally tagged LarA was less efficiently degraded by Lon than N-terminally tagged LarA (Fig. 1c, d) already indicated a role of the C-terminus in recognition by Lon. Sequence alignments with LarA orthologs from other α-proteobacteria revealed that this portion of the protein is highly conserved (Supplementary Fig. 5a), rich in

hydrophobic residues, a typical feature of Lon degron sequences[17], and has an terminal histidine like the well-characterized Sul20C degron[34]. Furthermore, an AlphaFold2 prediction of the LarA structure using ColabFold[35,36] indicated that the C-terminal region adopts an α-helical conformation followed by a short unstructured amino acid residue sequence (Supplementary Fig. 5b) that could potentially be involved in the interaction with Lon.

To experimentally investigate the role of the C-terminus of LarA in mediating the interaction with Lon, we first analysed a set of

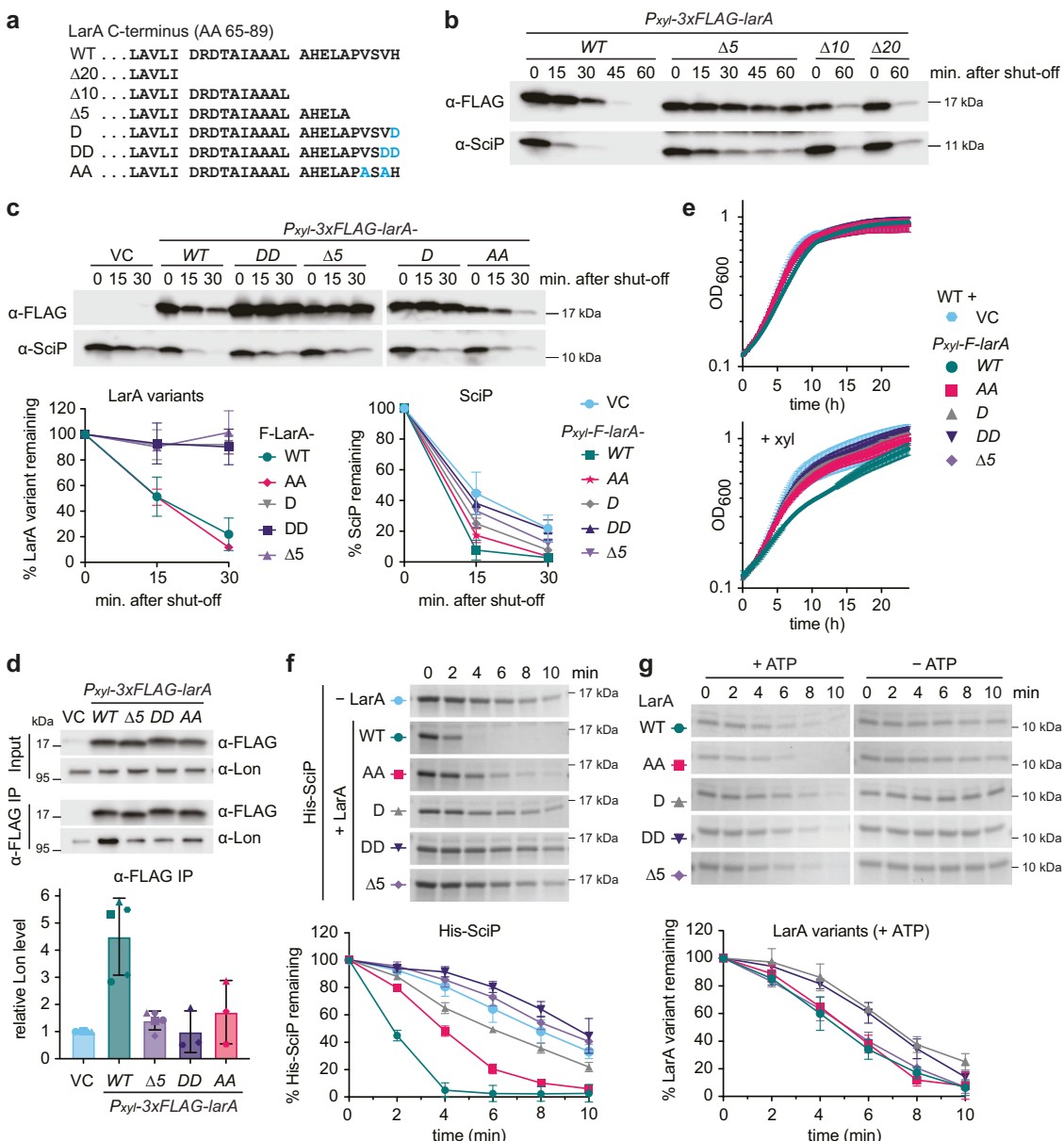

**Fig. 4 | LarA interacts with Lon via a C-terminal degron that is critical for Lon activation. a** Schematic representation of the C-terminal 25 amino acid (AA) residue sequence of LarA (AA 65–89) and the analysed truncation and point mutants. **b**, **c** In vivo stability of 3xFLAG-LarA (F-LarA) variants and SciP in wild type cells harbouring plasmids carrying the indicated constructs. The graphs in (**c**) show the mean values ± SD of F-LarA variant levels (left; *n* = 3 except *n* = 6 for F-LarA, biologically independent samples) and of SciP levels (right; *n* = 4 except *n* = 8 for *VC* and *F-larA*, biologically independent samples). **d** Immunoblots of 3xFLAG-tagged LarA (F-LarA) variants and Lon in lysates from WT cells (Input) and after α-FLAG immunoprecipitation. WT cells harboring either an empty vector (VC) or plasmids carrying *Pxyl-3xFLAG-larA* variants were grown for one hour with xylose to induce *F-larA* variant overexpression prior to cell lysis and IP. Lon levels after α-FLAG IP of F-LarA variants compared to the empty vector control (VC) are shown as mean values ± SD; *n* = 3 (*DD* and *AA*) or 5 (*VC, F-larA WT* and *Δ5*) biologically independent

samples from 3 or 4 independent IP experiments, respectively. **e** Growth curves of WT cells harbouring an empty vector (VC) or plasmids carrying the indicated constructs, grown under non-inducing conditions (upper panel) or *Pxyl*-inducing conditions (+xyl; lower panel). All growth curves display means ± SD; *n* = 4 or 2 (*Δ5*) biologically independent cultures. **f** In vitro degradation of His-SciP in the absence (−LarA) or in the presence of the indicated LarA variants. The reactions contained 0.25 μM Lon-His hexamer and 10 μM each of His-SciP and/or the respective LarA variant. Representatives gels are shown, quantifications show means ± SD; *n* = 3 or 5 (+LarA[WT], −LarA) independent experiments. **g** In vitro degradation of the LarA variants (10 μM) by Lon-His (0.25 μM) in the presence (+ATP) or absence (−ATP) of an ATP regeneration system. Gels are representatives of three independent experiments and the quantifications show mean values ± SD; *n* = 3 independent experiments. Source data are provided as a Source Data file.

truncated variants of F-LarA lacking the C-terminal 20, 10 and 5 amino acids (F-LarA[Δ20], F-LarA[Δ10], F-LarA[Δ5]; Fig. 4a). The Δ5 mutation removes merely part of the unstructured sequence, whereas the Δ10 mutation perturbs and the Δ20 mutation completely removes the α-helix (Supplementary Fig. 5b). In vivo degradation assays showed that the F-LarA[Δ5] variant was almost completely stabilised (Fig. 4b, c, Supplementary Fig. 6a, b). F-LarA[Δ10] and F-LarA[Δ20] were still degraded in vivo

(Supplementary Fig. 6a, b). However, degradation of these mutants was also observed in Δ*lon* cells, indicating that they are also degraded by other proteases (Supplementary Fig. 6a, b), probably due to structural changes that cause the exposure of additional degradation signals. Importantly, in contrast to full-length F-LarA, none of the three C-terminal F-LarA truncation mutants enhanced SciP degradation, as SciP was still readily detectable 60 min post translation shut-off

(Fig. 4b). Moreover, none of them led to growth inhibition when overexpressed (Supplementary Fig. 6c).

These data indicate a critical role of LarA's C-terminus in Lon binding and activation and are also highly consistent with our native MS data showing that the LarA$^{\Delta 5}$ variant does not form a complex with Lon (Fig. 3f). To seek additional support that the C-terminus of LarA mediates binding to Lon, we conducted a co-immunoprecipitation (Co-IP) experiment with an α-FLAG affinity matrix. Immunoprecipitation of F-LarA led to a significant enrichment of Lon in the elution fraction compared to an empty vector control, indicative of a stable interaction between F-LarA and Lon. By contrast, using F-LarA$^{\Delta 5}$ as the bait led to strongly reduced amounts of co-immunoprecipitated Lon (Fig. 4d), thus confirming that LarA binds Lon via its C-terminus.

We reasoned that the identities of the five C-terminal amino acid residues of LarA, some of which are highly conserved amongst LarA homologues (Supplementary Fig. 5a), are critical for the interaction and activation of Lon, and made more defined mutations in this region. In the first pair of F-LarA mutants the C-terminal one or two amino acid residues (H89 or V88 and H89) were replaced with one or two aspartates (D and DD), respectively (Fig. 4a), to change the nature of the five amino acid sequence from hydrophobic to negatively charged and to probe the importance of the presence of these residues. The F-LarA$^{H89D}$ (F-LarA$^D$) and F-LarA$^{V88D, H89D}$ (F-LarA$^{DD}$) mutants behaved in the in vivo degradation assays similar to the LarA$^{\Delta 5}$ mutant (Fig. 4c). They were clearly more stable than full length F-LarA and showed defects in stimulating SciP degradation (Fig. 4c). Moreover, in contrast to wild type F-LarA and similar to the LarA$^{\Delta 5}$ mutant, they showed reduced binding to Lon in the Co-IP assay (Fig. 4d) and failed to induce a growth defect when overexpressed (Fig. 4e). Thus, changing as little as the C-terminal histidine residue of LarA to a negatively charged aspartate residue is sufficient to disturb the regulatory interaction between LarA and Lon.

Next, we made more conservative mutations and replaced the two highly conserved valine residues, V86 and V88 (Supplementary Fig. 5a), with two alanine residues (Fig. 4a). We reasoned that these mutations would maintain the overall hydrophobic character of the C-terminal five amino acid sequence of LarA and would thus allow us to assess the contribution of the two bulky valine side chains to the LarA-dependent activation of Lon. In contrast to the F-LarA$^D$ and F-LarA$^{DD}$ mutants, this F-LarA$^{AA}$ mutant showed the same in vivo degradation kinetics as wild type LarA (Fig. 4c). Interestingly, despite being efficiently degraded by Lon, the LarA$^{AA}$ mutant displayed reduced binding to Lon (Fig. 4d), exhibited a partial defect in stimulating SciP degradation (Fig. 4c) and failed to induce a growth defect (Fig. 4e). This could suggest that the two valine residues at LarA's C-terminus make direct contacts with the allosteric regulatory site.

The defects of the different LarA variants in stimulating Lon-dependent degradation of SciP that we observed in vivo were also clearly visible when analysing their impact on SciP degradation in vitro. All mutants were defective in their ability to activate Lon-mediated proteolysis, with the LarA$^D$, LarA$^{DD}$ and LarA$^{\Delta 5}$ mutations showing the most drastic effects (Fig. 4f). Interestingly, in contrast to our in vivo data, the different purified LarA variants were all degraded by Lon themselves, in the case of the LarA$^{AA}$ and LarA$^{\Delta 5}$ variants with the same degradation kinetics as wild type LarA and in the case of the LarA$^D$ and LarA$^{DD}$ with an initial delay (Fig. 4g). This result may be explained by alternative degrons within LarA that are more readily recognized by Lon in the defined in vitro mixture. Importantly, although these alternative recognition signals facilitate Lon-dependent degradation of the LarA variants, they do not mediate allosteric activation of Lon, demonstrating that recognition via the C-terminal degron is a requirement for Lon activation.

Based on our mutational analysis we conclude that the unstructured five C-terminal amino acids of LarA compose a degron that is critical for LarA degradation in vivo and Lon activation by allowing access of LarA to an allosteric binding site on Lon. Based on our data,

we propose that upon engagement with Lon via this C-terminus, specific side chains in this region, including the two bulky valine and the basic histidine residues, make direct contacts with this allosteric binding site thereby inducing conformational changes of Lon that result in increased Lon activity.

## The C-terminal LarA degron is transferable

Based on the finding that the C-terminal five amino acid residues of LarA are critical for the interaction with Lon, we next investigated if this five amino acid-sequence is sufficient for mediating the degradation of a protein that is normally not a Lon substrate. For this, we transferred the 5, 10 or 20 C-terminal amino acids of LarA to the C-terminus of the I27 domain of human titin (titin-I27) (Fig. 5a), which can be unfolded by carboxymethylation of its cysteines (titin-I27$^{CM}$) and has served as a model for testing the autonomy of Lon-specific degron sequences[17,18]. Indeed, in contrast to untagged titin-I27$^{CM}$, which is not degraded by Lon[18] (Fig. 5b), all three resulting fusion proteins containing the LarA degron tags (titin-I27$^{CM}$-LarA5, titin-I27$^{CM}$-LarA10 and titin-I27 $^{CM}$-LarA20) showed robust degradation in the presence of ATP (Fig. 5b). Introducing the DD and D mutations to the titin-I27$^{CM}$-LarA5 variant largely abolished the degradation of the model substrate, supporting our previous finding that substituting as little as one C-terminal amino acid with a charged aspartate residue disrupts proper interaction with Lon (Fig. 5b). Introducing the AA mutations to titin-I27-LarA5 reduced degradation of the fusion protein as well, the effect was however milder compared to the effect of the DD and D mutations (Fig. 5b). These data demonstrate that the C-terminal degron of LarA is transferable and that only five amino acids are sufficient to target the non-substrate titin-I27$^{CM}$ for degradation by Lon.

Since the C-terminal residues of LarA not only mediate LarA degradation but also play a key role in the activation of Lon-dependent degradation, we tested the effect of titin-I27$^{CM}$-LarA20 on Lon-dependent His-SciP degradation. Analysing the effect of titin-I27$^{CM}$-LarA5 and titin-I27$^{CM}$-LarA10 was not possible as they cannot be separated from His-SciP by gel-electrophoresis. Titin-I27$^{CM}$-LarA20 was indeed able to stimulate in vitro His-SciP degradation (Fig. 5c). However, compared to the stimulatory effect of full length LarA (Fig. 3b), the observed effect was mild, indicating that in addition to the C-terminal amino acids other properties of LarA are required for its allosteric effects on Lon. This could either be additional specific residues within LarA or certain structural requirements that must be met to facilitate access to the allosteric site.

## LarA enhances the degradation of a variety of Lon substrates

So far, our data suggest that LarA acts as a dedicated allosteric activator of Lon that enhances Lon catalysis via a C-terminal binding sequence. Based on this, we hypothesised that LarA affects not only the degradation of SciP, but also of other Lon substrates. To test this hypothesis, we monitored the degradation of several other known Lon substrates in the absence and presence of LarA in vitro (Fig. 6a, b). Specifically, we tested the effect of LarA on the previously identified native *C. crescentus* Lon substrates CcrM and DnaA, as well as FliK-C, the C-terminal portion of FliK that contains the Lon-dependent degron[20,29,37]. Additionally, we assessed LarA-dependent effects on the degradation of the flagellar assembly protein FliX, a protein that was strongly enriched in the Lon trap (Fig. 1b) and thus might constitute a native Lon substrate, as well as of the unfolded model substrates β-casein and carboxymethylated titin-I27-β20 (titin-I27$^{CM}$-β20)[18,20]. Degradation rates of CcrM, FliK-C, FliX and β-casein were significantly increased to varying extents when LarA was present (Fig. 6a, b), demonstrating that LarA activates the degradation of a range of substrates. In the cases of FliK-C and FliX, we also tested if simultaneous incubation of LarA with either of these substrates positively affected the ATPase activity of Lon. Similar to what we observed when incubating Lon with a mix of LarA and SciP (Fig. 3e), ATPase

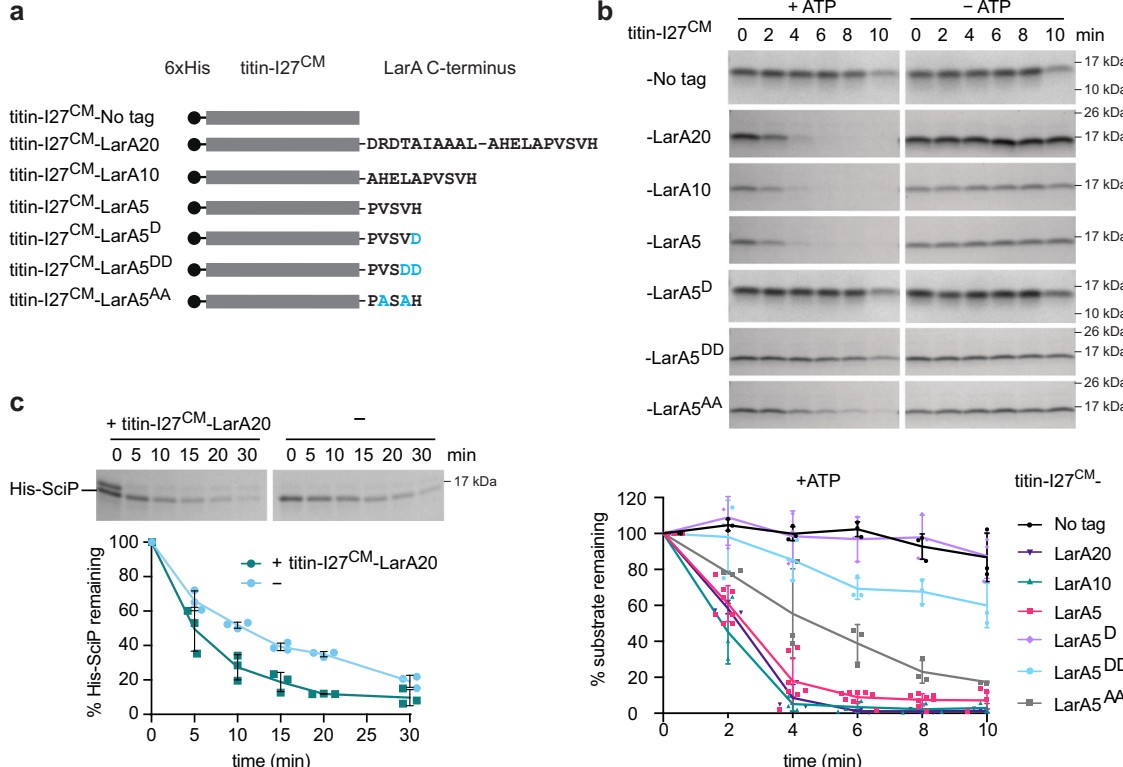

**Fig. 5 | The C-terminal LarA degron is transferable. a** Schematic representation of carboxymethylated (CM) titin-I27 without a tag (No tag) or with the C-terminal 20, 10 or 5 amino acid residues of LarA. The titin-I27-LarA5DD, LarA5D and titin-I27-LarA5AA mutants harbour the DD, D and AA substitutions, respectively, in the context of the titin-I27-LarA5 chimera. All titin-I27 variants harbour a 6xHis tag at the N-terminus. **b** In vitro degradation of unfolded titin-I27CM and of the unfolded titin-I27CM-LarA fusion constructs by Lon-His in the presence (+ATP) or absence (−ATP) of an ATP regeneration system. Quantifications show means ± SD of relative protein levels (normalised to Lon-His) from the +ATP condition; n = 3 or 4, 9 (LarA5) independent experiments. **c** In vitro degradation of His-SciP by Lon-His in the presence (+titin-I27CM-LarA20) or absence (−) of titin-I27CM-LarA20. Graph shows means ± SD of relative protein levels of His-SciP (normalised to Lon-His); n = 3 independent experiments. Source data are provided as a Source Data file.

activity was synergistically stimulated when LarA was co-incubated with FliX or FliK-C, and this effect was again dependent on the C-terminal 5 amino acid residues of LarA (Fig. 6c).

Interestingly, LarA did not significantly enhance degradation of the native substrate DnaA (Fig. 6a, b), and did not exhibit synergistic activation of ATPase activity of Lon when co-incubated with DnaA (Fig. 6c). LarA also did not stimulate in vitro degradation of the artificial substrate titin-I27CM-β20 (Fig. 6a, b)[17,18]. In this case, co-incubation with LarA resulted instead in a significant decrease in titin-I27CM-β20 degradation rate, suggesting that these two substrates may compete for the same or overlapping binding sites.

To study the effect of LarA on other Lon substrates in vivo, we assessed the effect of F-larA overexpression on FliK and DnaA, for which specific antibodies were available. In cells harbouring the full-length F-larA overexpression plasmid, FliK levels were drastically reduced in comparison to the vector control and the strain harbouring the plasmid for F-larAΔ5 overexpression, even without xylose addition (Fig. 6d), consistent with an activating effect of LarA on FliK degradation. By contrast, F-larA overexpression did not lower DnaA levels and also did not increase in vivo DnaA degradation (Supplementary Fig. 7a), which is consistent with our in vitro data. Prompted by our finding that the levels of FliK, a critical regulator of flagella hook length[29], were strongly downregulated in F-larA-overexpressing cells, we also tested the effect of F-larA overexpression on soft agar motility and observed a notable swimming defect. In contrast, no motility defect was observed when the F-LarAΔ5 mutant was overproduced (Supplementary Fig. 7b), again supporting our finding that the C-terminus of LarA is critical for LarA's regulatory effect on Lon.

Since SciP and FliK steady-state levels were strongly down-regulated in LarA-overexpressing cells as a consequence of their increased degradation (Supplementary Fig. 3b, Fig. 6d), we reasoned that other proteins may show a similar behaviour. Hence, to determine the group of Lon substrates affected by presence of LarA, we analysed proteome-wide protein abundances after 2 h of F-larA overexpression (Supplementary Dataset 2) and found that 65 proteins were significantly downregulated in F-LarA overexpressing cells compared to the vector control strain (Fig. 6e; Supplementary Dataset 2). These downregulated proteins included SciP as well as several other proteins that were previously described as validated or putative Lon substrates (e.g. FlgE, CheD) (Fig. 6e)[29]. Thus, the downregulation of these proteins in the F-larA-overexpression strain is likely due to increased degradation by Lon.

Our proteomics experiment also revealed a small group of proteins that were upregulated as a consequence of F-LarA over-expression (Fig. 6e). Interestingly, among the most upregulated proteins was Lon, an effect we could validate by Western blotting (Supplementary Fig. 7c). Together, our data demonstrate that LarA stimulates the degradation of a large and diverse group of Lon substrates, but also that the degradation of some substrates is unaffected or even negatively influenced by the presence of LarA.

## LarA accumulates in response to proteotoxic stress
Having defined the activating function of LarA on Lon, we investigated under which in vivo conditions LarA exerts this role. The gene encoding LarA is predicted to form an operon with the upstream located gene CCNA_03706[38], which encodes the small heat shock protein sHSP2 (Fig. 7a)[39]. According to previous RNA-sequencing and ChIP-sequencing

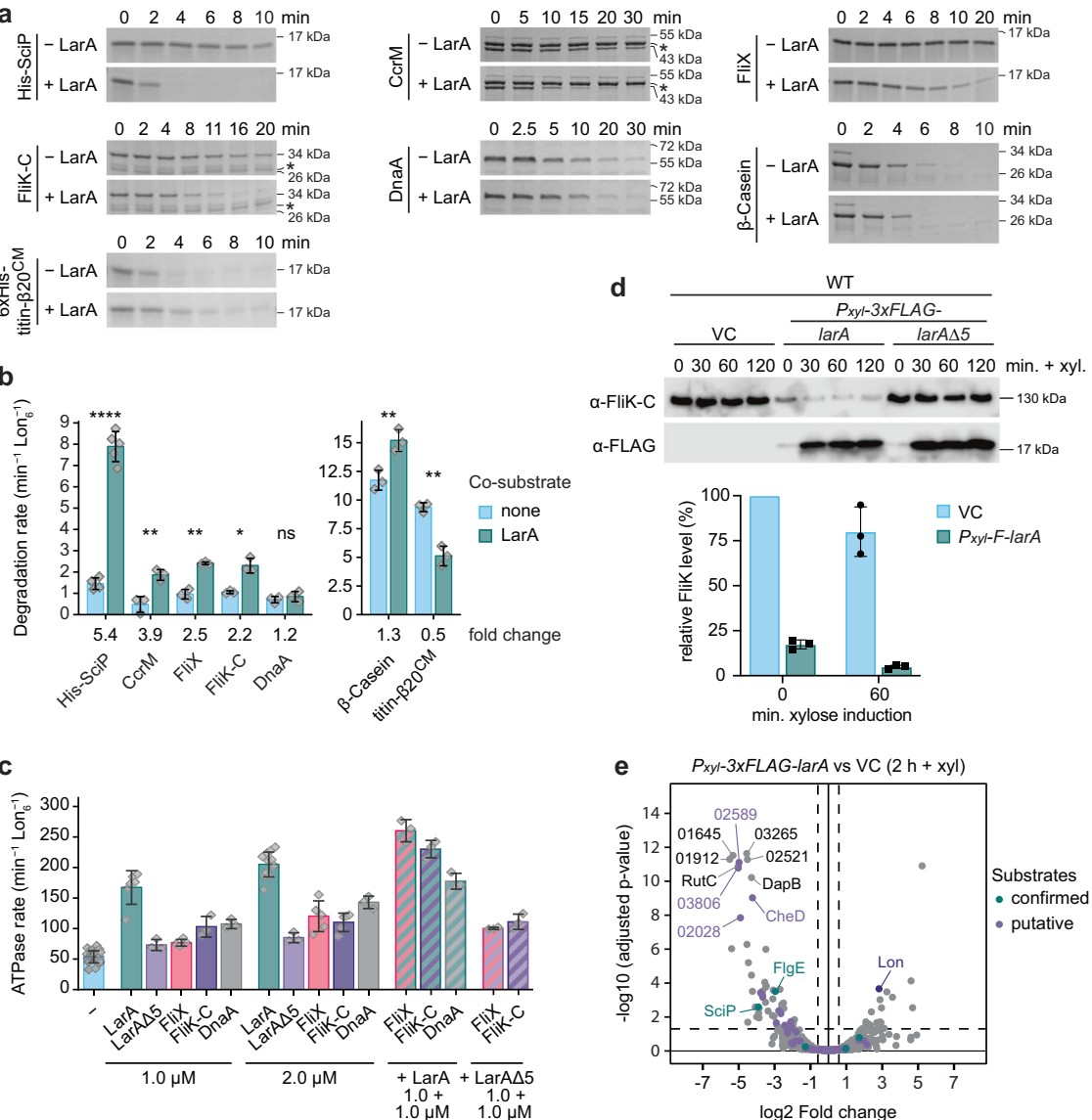

**Fig. 6 | LarA enhances the degradation of a variety of Lon substrates. a** In vitro degradation of various native and artificial Lon substrates in the absence (−LarA) or presence of LarA (+LarA). Substrates were used at the following concentrations: 3 μM LarA, 8 μM β-casein, 4 μM His-SciP, FliX, FliK-C, 2 μM CcrM, 1.5 μM DnaA, 3 μM 6xHis-titinI27-β20CM. Lon-His hexamer was added in all reactions at 0.125 μM, except for the reaction with 6xHis-titinI27-β20CM, in which 0.075 μM Lon-His hexamer was used. Representative gels are presented, $n = 3$, 4 (His-SciP − LarA), 5 (His-SciP + LarA) independent experiments. **b** Degradation rates of the Lon substrates shown in (**a**) in either the absence or presence of LarA. All reactions contained 0.05 μM Lon-His hexamer, 2 μM substrate and in case of co-degradations 2 μM LarA. The fold change of the LarA-dependent effect is indicated. Quantifications present the mean values ± SD, with sample sizes as reported in (**a**). Statistical significance was tested using an unpaired, two-sided Welch's t-test in R and the following $p$-values were obtained: His-SciP: $8.52 \times 10^{-6}$ ****; CcrM: 0.008 **; FliX: 0.005 **; FliK-C: 0.022 *; DnaA: 0.439 (ns); β-casein: 0.010 *; 6xHis-titinI27-β20CM (titin-β20CM): 0.006 **. **c** ATPase rate of Lon-His (0.05 μM) in the presence of different substrates. Bars for Lon alone (−) and when incubated with either LarA or LarA$^{Δ5}$ are shown for

comparison (reproduced from Fig. 3e). Labels below the bars indicate the substrate(s) and the used concentration(s). Bars and error bars represent the means ± SD; $n = 3$, 4 (1 μM FliX), 5 (2 μM FliX, 2 μM FliK-C), 6 (1 μM LarA), 12 (2 μM LarA), 36 (−) independent reactions. **d** Immunoblots of FliK and F-LarA levels in WT cells harbouring an empty vector (VC) or plasmids carrying $P_{xyl}$-3xFLAG-larA or $P_{xyl}$-3xFLAG-larA$^{Δ5}$. Quantifications show the means ± SD of FliK levels after 0 and 60 min of xylose addition compared to the empty vector control (VC); $n = 3$ biologically independent samples. **e** Volcano plot showing proteins affected by F-LarA overexpression. WT cells harbouring either an empty vector (VC) or the plasmid carrying $P_{xyl}$-3xFLAG-larA were grown for 2 h with xylose to induce $F$-larA overexpression. Lon, the confirmed substrates SciP and FlgE (green label) as well as the ten most significantly changed proteins are indicated (previously reported putative substrates[29] are labeled in purple). Analysis of three biological replicates ($n = 3$) using "Differential Enrichment analysis of Proteomics data" (DEP) of bioconductor is shown (see Methods and Supplementary Dataset 2 for details and raw values). Source data are provided as a Source Data file.

data[40,41], this operon is directly regulated by the heat shock sigma factor $\sigma^{32}$ and transcriptionally induced under conditions activating the heat shock response, for example during depletion of the chaperone DnaK, a negative regulator of $\sigma^{32}$ [41,42] (Fig. 7b). Consistent with these data, we observed a gradual accumulation of LarA following depletion of DnaK when analysing LarA levels by Western blotting using an antibody

specific for endogenous LarA (Fig. 7c). Interestingly, in cells lacking Lon, LarA was detectable even in the presence of DnaK and accumulated to much higher levels after DnaK depletion (Fig. 7c). This suggests that *larA* is expressed under normal growth conditions, but is efficiently degraded by Lon, consistent with our finding that LarA is a good Lon substrate and stabilised in Δ*lon* cells (Fig. 1c–f).

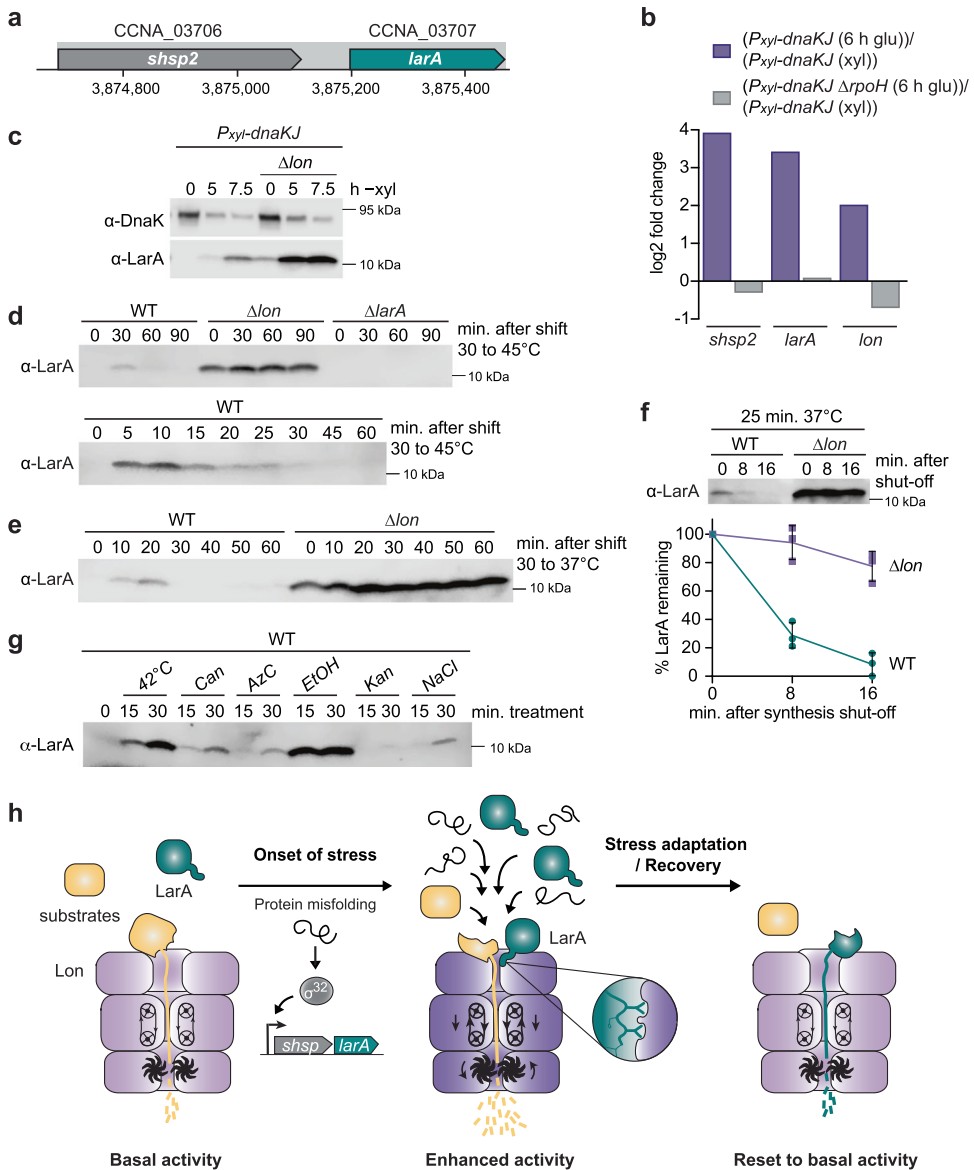

**Fig. 7 | LarA accumulates in response to proteotoxic stress. a** Schematic representation of the operon containing CCNA_03706 (*shsp2*) and CCNA_03707 (*larA*). **b** Changes in *shsp2*, *larA* and *lon* expression induced by 6 h of DnaKJ depletion (6 h glu) compared to the non-depleted condition (xyl) in an otherwise WT strain (purple) or in a strain lacking the heat shock sigma factor σ32 (Δ*rpoH*) (grey). The quantifications are based on previously published RNA-sequencing data[41]. **c** Immunoblots showing DnaK and LarA levels in the $P_{xyl}$-*dnaKJ* depletion strain and the $P_{xyl}$-*dnaKJ* Δ*lon* strain. Samples were taken at the indicated time points after change of growth medium from non-depleting with xylose to DnaKJ depleting medium without xylose (−xyl). Representative data are shown; *n* = 2 biologically independent samples. **d** Immunoblots showing LarA levels in WT, Δ*lon* and Δ*larA* strains. Samples were taken before (0) and at the indicated time points after shifting the cultures from 30 °C to 45 °C. Representative data are shown; *n* = 3 biologically independent samples. **e** Immunoblot showing LarA levels in WT and

Δ*lon* strains after shifting the cultures from 30 °C to 37 °C. Representative data are shown; *n* = 3 biologically independent samples. **f** In vivo stability of LarA in WT and Δ*lon* cells. Cultures were shifted from 30 °C to 37 °C and incubated for 25 min prior to protein synthesis shut-off at 0 min. Quantifications show LarA levels as mean values ± SD, *n* = 3 biologically independent samples. **g** Immunoblot showing induction of LarA levels in WT cells before (0 min) and after treatment (15 and 30 min) with various stress conditions inducing proteotoxic stress; heat stress at 42 °C (42 °C), addition of L-canavanine (Can; 250 µg/ml final conc.), addition of azetidine-2-carboxylate (AzC; 5 mM final), ethanol (EtOH; 5% final), kanamycin (Kan; 0.1125 µg/ml final), sodium chloride (NaCl; 85 mM final). Representative data are shown; *n* = 4 biologically independent samples. **h** Model of LarA-dependent activation of Lon at the onset of proteotoxic stress. See main text (Discussion) for a detailed description. Source data are provided as a Source Data file.

Next, to analyse how LarA abundance is affected by temperature changes, we monitored LarA protein levels under optimal conditions and following a shift to heat stress temperatures. In wild type cells grown at 30 °C (corresponding to *t* = 0), LarA levels were below detection limit, whereas Δ*lon* cells showed again readily detectable levels of LarA under the same condition (Fig. 7d). Upon shift to 45 °C, LarA levels strongly increased within 10 min in the wild type, which was followed by a reduction in steady-state levels during continued incubation at 45 °C (Fig. 7d). A similar pattern was observed under mild

heat shock conditions when shifting cells from 30 °C to 37 °C (Fig. 7e). In the Δ*lon* mutant, the temperature-dependent regulation of LarA levels was severely perturbed with continuously high levels of LarA throughout the experiment (Fig. 7d, e). Monitoring the stability of native LarA with the LarA-specific antibody revealed that LarA was rapidly degraded in the wild type but nearly completely stabilised in Δ*lon* cells (Fig. 7f), indicating that the fast turnover of LarA through Lon-mediated proteolysis is required to temporally restrict LarA accumulation. We also analysed LarA accumulation under several

other proteotoxic conditions and observed different degrees of LarA upregulation in the presence of sublethal concentrations of ethanol (EtOH), sodium chloride, kanamycin or the amino acid analogs canavanine (Can) and azetidine-2-carboxylic acid (AzC) (Fig. 7g). Interestingly, while EtOH stress resulted in rapid and transient upregulation of LarA, similar to the response to heat stress, Can and AzC treatment caused an accumulation of LarA that persisted for a longer period of time (Supplementary Fig. 8a). Having identified proteotoxic stress conditions, in which LarA is upregulated, we also assessed the growth phenotype of a ΔlarA mutant under these conditions in comparison to Δlon and wild type cells. While the Δlon mutant exhibited a mild growth defect even in the absence of stress, we did not detect an obvious growth phenotype of the ΔlarA strain under the chosen conditions (Supplementary Fig. 8b–i). This suggests that the absence of LarA-mediated Lon activation may be compensated by other components of the highly redundant proteostasis network. Alternatively, LarA might be particularly needed under other unknown stress conditions. Together, our data show that LarA levels are upregulated in response to various proteotoxic stress conditions and that the transient accumulation of LarA at the onset of heat stress is achieved by a combination of $\sigma^{32}$-dependent transcription and Lon-mediated proteolysis of LarA.

## Discussion

### A model of LarA-dependent Lon regulation at the onset of stress

In this study, we describe the discovery of a α-proteobacterial heat shock protein, LarA (previously CCNA_03707), and its characterisation as a regulator of the Lon protease. Based on our data we propose the following model of LarA-dependent regulation of Lon activity (Fig. 7h). In the absence of stress, LarA levels are low due to basal transcription of larA and constitutive Lon-dependent degradation of LarA. Hence, Lon operates at basal activity. Under conditions that trigger $\sigma^{32}$-dependent heat shock gene expression, e.g., proteotoxic stress caused by temperature upshift, larA expression is induced and LarA accumulates. As LarA levels rise, it interacts with Lon via its accessible C-terminus. Upon initial engagement with Lon, specific amino acid residues within this C-terminal degron interact with an allosteric regulatory site. This site is likely located in the N-terminal domain of Lon, which was previously shown to be required for the recognition of degron-tagged substrates[43]. The binding of LarA induces conformational changes of Lon leading to the enhancement of the ATPase and proteolytic activities of Lon. Activated Lon degrades a broad range of substrates more efficiently, including both folded and unfolded proteins. The LarA-dependent Lon activation in combination with the previously reported activation of Lon by unfolded proteins[20], likely guarantees continued degradation of specific substrates when the proteolysis demand is high. Upon restoration of protein homeostasis, $\sigma^{32}$-dependent larA expression ceases again, which leads in combination with fast Lon-dependent proteolysis to rapid downregulation of LarA. This reduction in LarA levels resets Lon activity to basal levels, which is critical to avoid toxic consequences of continued high Lon activity, which not only targets un- or misfolded polypeptides but also functional native proteins. The fast and temporally restricted upregulation of LarA may help Lon to meet the proteolytic demand during the initial stages of the stress response.

### Molecular mechanism of LarA-dependent Lon activation

By what molecular mechanism does the interaction between LarA and Lon enhance Lon proteolysis? Our in vitro and in vivo data suggest that LarA binding to a distinct allosteric site induces conformational changes in Lon that lead to the observed robust and strong changes in ATPase and proteolytic activity. These conformational changes also enhance the affinity of Lon to specific substrates, likely by exposing residues involved in substrate binding. While LarA may merely induce local conformational changes within a pre-assembled hexamer, it is tempting to speculate that LarA affects the quaternary structure of Lon in a more drastic manner. Recent structural studies suggest that Lon hexamers, like other AAA+ proteases, shift between distinct assembly states[33,44]. In this model, Lon adopts an open conformation with auto-inhibited proteolytic active sites in the absence of a substrate. Upon substrate binding, Lon undergoes a reorganization to a closed conformation with active protease sites. Accordingly, LarA might stabilise Lon in the closed conformation thereby increasing its activity towards other substrates. In an alternative scenario, LarA binding to Lon affects its oligomeric state. Studies in E. coli suggested that Lon likely equilibrates, by means of N-domain interactions, between a hexameric and a dodecameric assembly state with altered activity and substrate preference[45]. Determining the structure of Lon bound to LarA will provide insights into these questions.

Intriguingly, while the degradation of a large group of different Lon substrates is enhanced by LarA, the degradation of two Lon substrates, DnaA and unfolded titin-I27$^{CM}$-β20 (Fig. 6), was not or even negatively affected by LarA. The differences in responsiveness to the allosteric activation by LarA likely provides a means to globally reprogram Lon selectivity, facilitating large scale remodeling of the proteome content in response to changing conditions.

### Phylogenetically distinct Lon regulators

The existence of a regulatory protein that stabilises Lon in an activated conformation during heat shock or other stresses to facilitate enhanced proteolysis was previously proposed[18], but remained elusive. Intriguingly, our data suggest that LarA is such a dedicated Lon activity regulator. Given that LarA is specific to a group of α-proteobacteria, other species have likely evolved other proteins that carry out this task. For example, the previously described Lon specificity-enhancing factor and substrate HspQ, a YccV protein family member, shows functional similarities to LarA. Similar to LarA, the expression of hspQ is heat shock-induced[46], it activates Lon through a C-terminal binding sequence and promotes the degradation of certain Lon substrates[24]. Despite these similarities, there are several key differences between these two Lon regulators. In contrast to LarA, simultaneous incubation of HspQ with another substrate did not result in synergistic stimulation of ATPase activity, indicating mechanistic differences in the way these proteins act on Lon[24]. Furthermore, unlike HspQ, LarA is co-regulated with a small heat shock protein, suggesting a functional relationship between LarA and this sHSP. Finally, in contrast to HspQ, which was previously described as a hemimethylated DNA binding protein[47] and has, in addition to its stimulatory function on Lon, also regulatory functions on ClpSAP[25], our data suggest that LarA is a dedicated Lon regulator. Overexpression of larA resulted in a strong growth defect in a strictly Lon-dependent manner (Fig. 2), indicating that the main function of LarA is its regulatory role on Lon.

In conclusion, our study suggests that different phylogenetic groups of bacteria have evolved distinct Lon regulators which can have similar functions. It is likely that a plethora of additional Lon regulators exist in other organisms. Identifying those will be an important task for the future to gain deeper insights into the precise regulation of the activity of Lon and other AAA+ enzymes.

## Methods

### Strains, plasmids and oligonucleotides

All bacterial strains, plasmids and oligonucleotides used in this study are listed in Supplemental Supplementary Tables 3, 4 and 5, respectively.

### Plasmid constructions

**Plasmids for protein expression.** Plasmids for protein expression of fliX (pMF58-c4), larA (pMF65-c5), larA-V86A-V88A (pMF82), larA-H89D (pMF81-c2), larA-V88D-H89D (pMF88-c4), larAΔ5 (pMF89-c2) were constructed as follows.

The coding sequences for FliX and LarA (CCNA_03707) were amplified from genomic DNA using primer pairs oMJF47/oMJF48 and oMJF67/oMJF68, respectively, and cloned into pSUMO-YHRC by Gibson Assembly to generate pMF58-c4 and pMF65-c5.

Expression vectors for LarA mutants were created by amplifying the LarA coding sequence from pMF65-c5 using primers oMJF67/oMJF97 (pMF82), oMJF67/oMJF96 (pMF81-c2), oMJF67/oMJF106 (pMF88-c4) or oMJF67/oMJF107 (pMF89-c2) and cloning into pSUMO-YHRC by Gibson Assembly.

Expression vector for Lon with a mutated Walker B motif (LonE420Q or Lon$^{EQ}$) was created by amplifying two overlapping fragments of the coding sequence from pBAD33-ccLon introducing the Walker B mutation using primers oMJF90/oMJF94 and oMJF95/oMJF91. The fragments were separated on an agarose gel and extracted using the QIAquick Gel extraction kit (#28706 × 4) and 20 ng of each fragment was used for a fusion PCR using primers oMJF90/oMJF91. Subsequently the mutated Lon coding sequence was cloned into pSUMO-YHRC by Gibson Assembly.

Additional details of the cloning procedure can be found in Omnus et al.[29]. Successful subcloning was confirmed by Sanger sequencing using primers T7 and T7 terminator.

**Expression plasmids for purification of 6xHis-titinI27-LarA variants.** Double-stranded DNA fragments encoding the sequences of the 5, 10 or 20 amino acids of the C-terminal degron of LarA flanked by BamHI and SpeI restriction sites were created by annealing oligonucleotide pairs OAK057/OAK058, OAK059/OAK060 and OAK061/OAK062 respectively. The resulting DNA fragments as well as the pSH21 plasmid containing *6xHis-titinI27-β20*[48] were digested using BamHI-HF and SpeI-HF and subsequently gel-purified (using QIAquick gel extraction kit, Qiagen). Following dephosphorylation of the cut plasmid using Antarctic phosphatase, the respective DNA fragments were inserted using T4 DNA ligase at 16 °C overnight. DH5α competent cells were transformed with the ligation reaction and plated on ampicillin-containing LB plates. The *6xHis-titinI27* constructs with no added tag and *larA5* mutant variants (*larA5-H89D, larA5-V88D-H89D* and *larA5-V86A-V88A*) were constructed in the same manner, but by using oligonucleotide pairs OAK085/OAK086, OAK077/OAK078 OAK079/OAK080 and OAK081/OAK082, respectively.

**Replicating plasmids**
**pDJO26 (pBX-MCS-2 carrying *lon$^{WT}$-Twin-Strep-tag*).** *lon* was amplified using primers oDJO16 and oDJO18 as well as pML1716-*lon* (KJ600) as a template. The resulting PCR product was purified and DpnI-treated. To add the sequence coding for a Twin-Strep-tag (AS-WSHPQFEK-(GGGS)$_3$-WSHPQFEK-GA) in frame to the 3' end of *lon*, two additional PCRs were conducted. First, the *lon* PCR product was used as a template for a PCR with primers oDJO16 and oDJO21. The resulting PCR product was again gel-purified and then used as template for PCR with primers oDJO16 and oDJO22. The final PCR product was gel-purified and joined with PCR-amplified (primers oDJO13 and oDJO15), DpnI-treated, pBX-MCS-2 by Gibson assembly[49].

**pDJO40 (pBX-MCS-2 carrying *lon$^{TRAP}$-Twin-Strep-tag*).** Plasmid pDJO26 (pBX-MCS-2 carrying *lon$^{WT}$-Twin-Strep-tag*) was used as template for two PCR reactions, a PCR with primers oDJO16 and oDJO24 and a second PCR with primers oDJO23 and oDJO20 to introduce the TRAP mutation (S674A) into *lon*. The resulting PCR products were gel purified and joined with PCR-amplified (primers oDJO13 and oDJO15), DpnI-treated, pBX-MCS-2 by Gibson assembly.

**pDJO305 (pBX-MCS-4 containing *P$_{xyl}$-larA-3xFLAG*).** *larA* (CCNA_03707) was amplified with primers oDJO171 and oDJO172 using chromosomal *C. crescentus* NA1000 DNA as template and cloned into NdeI-cut pDJO145 using Gibson assembly.

**pDJO307 (pBX-MCS-4 containing *P$_{xyl}$−3xFLAG-larA*).** *larA* (CCNA_03707) was amplified with primers oDJO173 and oDJO174 using chromosomal *C. crescentus* NA1000 DNA as template and cloned into KpnI-cut pDJO145 using Gibson assembly.

**pDJO374 (pBX-MCS-4 containing *P$_{xyl}$−3xFLAG-larAΔ5*).** The sequence coding for *larA* except the C-terminal 5 amino acids (*larAΔ5*) was amplified with primers oDJO173 and oDJO182 using chromosomal DNA as template, purified, and cloned into KpnI-cut pDJO145 using Gibson assembly.

**pDJO377 (pBX-MCS-4 containing *P$_{xyl}$−3xFLAG-larAΔ10*).** The sequence coding for *larA* except the C-terminal 10 amino acids (*larAΔ10*) was amplified with primers oDJO173 and oDJO183 using chromosomal DNA as template, purified, and cloned into KpnI-cut pDJO145 using Gibson assembly.

**pDJO380 (pBX-MCS-4 containing *P$_{xyl}$−3xFLAG-larAΔ20*).** The sequence coding for *larA* except the C-terminal 20 amino acids (*larAΔ20*) was amplified with primers oDJO173 and oDJO184 using chromosomal DNA as template, purified, and cloned into KpnI-cut pDJO145 using Gibson assembly.

**pDJO451 (pBX-MCS-4 containing *P$_{xyl}$−3xFLAG-larA-H89D*).** *larA* was amplified with primers oDJO173 and oDJO197 to introduce the H89D mutation using chromosomal *C. crescentus* NA1000 DNA as template and cloned into KpnI-cut pDJO145 using Gibson assembly.

**pDJO460 (pBX-MCS-4 containing *P$_{xyl}$−3xFLAG-larA-V88D-H89D*).** *larA* was amplified with primers oDJO173 and oDJO198 to introduce the V88D and H89D mutations using chromosomal *C. crescentus* NA1000 DNA as template and cloned into KpnI-cut pDJO145 using Gibson assembly.

**pDJO461 (pBX-MCS-4 containing *P$_{xyl}$−3xFLAG-larA-V86A-V88A*).** *larA* was amplified with primers oDJO173 and oDJO199 to introduce the V86A and V88A mutations using chromosomal *C. crescentus* NA1000 DNA as template and cloned into KpnI-cut pDJO145 using Gibson assembly.

**Integrating plasmids**
**pDJO67 (pXCHYN-2 containing *lon$^{WT}$-Twin-Strep-tag*).** *lon$^{WT}$-Twin-Strep-tag* was amplified using primers OFS932 and oDJO25 and pDJO26 as template. The resulting PCR product was gel purified and assembled into NdeI and SacI-cut and gel purified plasmid pXCHYN-2 by Gibson assembly.

**pDJO70 (pXCHYN-2 containing *lon$^{TRAP}$-Twin-Strep-tag*).** *lon$^{TRAP}$-Twin-Strep-tag* was amplified using primers OFS932 and oDJO25 and pDJO40 as template. The resulting PCR product was gel purified and assembled into NdeI and SacI-cut and gel purified plasmid pXCHYN-2 by Gibson assembly.

**pDJO404 (pNPTS138-*UHR-tet$^R$-DHR* (*larA*) to generate a *tet$^R$*-marked deletion of *larA* (CCNA_03707).** A fragment containing the upstream homology region (UHR) comprising the 554 bp upstream and the first 15 bp of CCNA_03707 was amplified from chromosomal *C. crescentus* NA1000 DNA using primers oDJO185 and oDJO186. The downstream homology region (DHR) encompassing the last 27 bp and the 571 bp downstream of CCNA_03707 was amplified with primers oDJO187 and oDJO188. The homology regions were assembled together with an OFS25/26-amplified tetR-cassette into EcoRI and HindIII-cut pNTPS138 by Gibson assembly.

**Bacterial strain construction**
**KJ1066 (DJO002; Δlon P$_{xyl}$-lon$^{WT}$-Twin-Strep-tag).** The *lon$^{WT}$-Twin-Strep-tag* construct was introduced into the Δ*lon* strain LS2382 at the

*xylX* locus by transformation with plasmid pDJO67. Briefly, transformants were selected on kanamycin plates, single colonies were obtained and the successful integration was confirmed by colony PCR using primers RecUni-1 and RecXyl-2.

**KJ1067 (DJO003; Δ*lon* P*xyl*-*lon*TRAP-Twin-Strep-tag).** The *lon*TRAP-Twin-Strep-tag construct was introduced into the Δ*lon* strain LS2382 at the *xylX* locus by transformation with plasmid pDJO70. Briefly, transformants were selected on kanamycin plates, single colonies were obtained and the successful integration was confirmed by colony PCR using primers RecUni-1 and RecXyl-2.

**KJ1068 (DJO004; WT P*xyl*-*lon*WT-Twin-Strep-tag).** The *lon*WT-Twin-Strep-tag construct was introduced into the WT strain NA1000 at the *xylX* locus by transformation with plasmid pDJO67. Briefly, transformants were selected on kanamycin plates, single colonies were obtained and the successful integration was confirmed by colony PCR using primers RecUni-1 and RecXyl-2.

**KJ1069 (DJO005; WT P*xyl*-*lon*TRAP-Twin-Strep-tag).** The *lon*TRAP-Twin-Strep-tag construct was introduced into WT strain NA1000 at the *xylX* locus by transformation with plasmid pDJO70. Briefly, transformants were selected on kanamycin plates, single colonies were obtained and the successful integration was confirmed by colony PCR using primers RecUni-1 and RecXyl-2.

**KJ1070 (DJO006; Δ*larA*).** The Δ*larA* deletion was introduced into the WT strain NA1000 by two-step recombination[50] after transformation with plasmid pDJO404. Briefly, transformants were selected on kanamycin and tetracycline plates, single colonies were grown overnight in PYE and plated on PYE tetracycline plates containing sucrose. Single sucrose-resistant colonies were subsequently screened for kanamycin sensitivity and tetracycline resistance and the *larA* knockout was confirmed by colony PCR using primers oDJO193 and oDJO194.

*C. crescentus* strains carrying replicating plasmids were created by transforming the plasmids into the respective strain background by electroporation.

## Bacterial growth conditions
*C. crescentus* strains were routinely grown at 30 °C on solid PYE medium agar or in liquid PYE medium while shaking at 200 rpm and, if necessary, regularly diluted to assure growth in the exponential phase. Growth media were supplemented with xylose (0.3% final) when indicated. Antibiotics in selective media were used at following concentration (concentration in liquid/solid media as μg/ml): gentamycin (0.625/5), chloramphenicol (1/1), oxytetracycline (1/2), kanamycin (5/25). Experiments were generally performed in the absence of antibiotic when using strains in which the resistance cassette was integrated into the chromosome.

*E. coli* strains for cloning purposes were grown on LB medium plates or LB liquid medium at 37 °C. *E. coli* strains for protein expression were grown using LB medium (ER2566, BL21(DE3)/pLysS) or LBON/2xYTON (BL21-SI/pCodonPlus). If necessary, media was supplemented with antibiotics at following concentrations (concentration in liquid/solid media as μg/ml): ampicillin (100/50), gentamycin (15/20), kanamycin (30/50), oxytetracycline (12/12), chloramphenicol (20/40).

## Recording of growth curves using a plate reader
Cultures were diluted to an OD$_{600}$ of 0.05, split in two aliquots and xylose was added to one of the aliquots to induce expression of Lon$^{WT}$-Twin-Strep-tag or Lon$^{TRAP}$-Twin-Strep-tag as well as of 3xFLAG-LarA and 3xFLAG-LarA variants as indicated. Subsequently, 200 μL volumes were transferred into sterile 96 well plates. Cells were cultured at 30 °C and the OD$_{600}$ was measured every 10 min using a Spark microplate reader (Tecan) with TECAN SparkControl magellan 2.2 software.

## Motility assay
To assess motility, strains were grown in PYE media supplemented with gentamycin to maintain replicating plasmids, and cultures were diluted to an OD$_{600}$ of 0.1. Subsequently, 1 μL of each sample was injected about 2 mm vertically into PYE agar plates (0.35%) supplemented with gentamycin or gentamycin and xylose when indicated, using a pipette. The plates were incubated at 30 °C and pictures were taken with the setting Blots: Colorimetric using a ChemiDoc (Bio-Rad ImageLab 5.1/6.1.0).

## Proteomics-based identification of Lon-bound proteins
To identify Lon-bound proteins, the expression of either Lon$^{WT}$ or Lon$^{TRAP}$ constructs harbouring a C-terminal Twin-Strep-tag integrated at the *xylX* locus of Δ*lon* cells were induced by xylose addition to exponentially growing cultures for 2 h. Subsequently, Lon$^{WT}$-Twin-Strep-tag and Lon$^{TRAP}$-Twin-Strep-tag were purified using the Twin-Strep-tag purification system as described below. As control (No Lon), a culture of Δ*lon* cells not expressing any Twin-Strep-tagged construct was processed in parallel. The expression and purification procedures were repeated to obtain in total two independent biological replicates. For replicate 1, elution fractions 5 (E5) and for replicate 2, elution fractions 5 and 6 as well as 7 for Lon$^{TRAP}$ only, were submitted for quantitative and qualitative protein content determination.

Protein digestion, TMT10 plex isobaric labeling and the mass spectrometrical analysis was performed by the Clinical Proteomics Mass Spectrometry facility, Karolinska Institute/Karolinska University Hospital/Science for Life Laboratory. Briefly, denatured samples were subjected to Single-Pot Solid-Phase-enhanced Sample Preparation (SP3) procedure[51]. The resulting peptide mixtures were labelled with isobaric TMT-tags according to the manufacturer's protocol (Thermo Scientific). Online LC-MS was performed using a Dionex UltiMate™ 3000 RSLCnano System coupled to a Q-Exactive HF mass spectrometer (Thermo Scientific). Samples were trapped on a C18 guard desalting column and separated on a 50 cm long C18 column. The MS raw files were searched using Sequest-Percolator (18-05-2018) under the software platform Proteome Discoverer 1.4 (Thermo Scientific) against *C. crescentus* Uniprot database (19-12-2016) and filtered to a 1% FDR cut off.

For downstream analysis (Supplementary Dataset 1), values of elution fractions E5 were used, except for Lon$^{TRAP}$ of replicate 2 where elution fraction E6 was chosen instead due to the Lon level detected in this fraction, which was most similar to the other Lon-construct containing samples selected. Proteins were considered to be specifically enriched in the Lon$^{TRAP}$ fraction if they were only detected in Lon$^{TRAP}$ sample(s) or more abundant in Lon$^{TRAP}$ sample(s) than in the control and the Lon$^{WT}$ samples. In Fig. 1b only the proteins are displayed that were detected in both replicates and for which the average value for Lon$^{TRAP}$ was higher than for no Lon control and Lon$^{WT}$.

## Quantitative proteomics analysis of LarA overexpressing cells
To analyse proteome-wide changes after LarA overexpression, cultures of wild type (WT) cells either harbouring an empty vector (vector control, VC) or the plasmid for xylose-inducible overexpression of 3xFLAG-tagged LarA (P*xyl*–3xFLAG-*larA*) were harvested by centrifugation. For the non-induced cultures, two biological replicates were harvested as well as three biological replicates each after 2 and 4 h of xylose addition. Cell pellets were washed using cold ddH2O and stored at −80 °C. Protein digestion, TMT isobaric labeling and the mass spectrometry analysis were performed by the Clinical Proteomics Mass Spectrometry facility, Karolinska Institute/Karolinska University Hospital/Science for Life Laboratory by similar procedures as described above, but using a TMTpro 16 plex labeling procedure.

Analysis of the quantitative proteomics results was performed using R version 4.2.0 in RStudio (http://www.rstudio.com/). Normalised TMT ratios that express the relative abundance of each protein in

each sample were analysed with the Differential Enrichment analysis of Proteomics data (DEP) package[52] from Bioconductor according to manual to obtain and visualize differential protein abundances. To test for differentially expressed proteins, the threshold of the adjusted p-value and the log2 fold change in abundance was set to 0.05 and 1.5, respectively (see Supplementary Dataset 2). The quality of the data and the analysis was assessed using various control functions as PCA, MeanSDPlots and others suggested in the manual of the DEP package. Finally, volcano plots of the comparison between experimental groups F-LarA overexpression and vector control after 2 h xylose induction were created (Fig. 6e).

### Immunoblot analysis

For whole cell extract analysis, 1 mL culture samples were collected after the indicated treatments and time points, and cell pellets were obtained by centrifugation. Cell pellets were resuspended in appropriate amounts of 1x SDS sample buffer, to ensure normalisation of the samples by units $OD_{600}$ of the cultures. Samples acquired from Twin-Strep-tag purification elution fractions were processed by adding appropriate amounts of concentrated sample buffer/volume. Samples were boiled at 98 °C for 10 min and frozen at −20 °C until further use. The thawed samples were separated by SDS-PAGE using Mini-PRO-TEAN® TGX Stain-Free™ gels (usually 4–20%, Bio-Rad), and subsequently transferred to nitrocellulose membranes by a semi-dry blotting procedure as per manufacturer guidelines. The protein gels and membranes were imaged using a Gel Doc Imager before and after the transfer, respectively, to assess equal loading of total protein as well as the quality of the transfer.

Membranes were blocked for 1 h at room temperature or overnight at 4 °C in 10% skim milk powder in TBS-Tween (TBST) and protein levels were detected using the following primary antibodies and dilutions in 3% skim milk powder in TBST: anti-DnaA[53] (1:5000), anti-Lon[45] (1:10,000), anti-FLAG M2 antibody (1:1000; Sigma, #F1804-1MG), anti-SciP[31] (1:2000), anti-FliK-C[29] (1:300), anti-LarA (this study, 1:250). Secondary antibodies, 1:5000 dilutions of anti-rabbit or anti-mouse HRP-conjugated antibodies (Thermo Fisher Scientific), and Super-Signal Femto West (Thermo Fisher Scientific) were used to detect primary antibody binding. Immunoblots were scanned using a Chemidoc system (Bio-Rad) with BioRad ImageLab 5.1/6.1.0 software or a LI-COR Odyssey Fc system with LI-COR Image Studio 5.2.5 software. Relative signal intensities were quantified using ImageJ or the Image Lab software package (BioRad ImageLab 5.1/6.1.0).

### In vivo degradation assay

To analyse protein stability in vivo protein synthesis was shut-off by addition of chloramphenicol (100 μg/mL final). Samples were subsequently taken at the indicated time points and snap frozen in liquid nitrogen before preparation for analysis by immunoblotting. When needed, the expression of FLAG-tagged LarA and its derivatives was induced by addition of xylose to the cultures, usually 1 h prior to chloramphenicol addition.

### α-FLAG co-immunoprecipitation (Co-IP)

Cells harboring the vector control (pBX-MCS-4) or plasmids carrying Pxyl-F-larA variants (WT, D5, DD, AA) were grown in PYE with gentamycin and expression was induced by addition of xylose (final 0.3%) for 1 h. 1 OD of cells was harvested by centrifugation (5 min at 6000 × g). Cell pellets were resuspended in 170 μl lysis buffer, i.e., BugBuster Master Mix (Millipore) containing protease inhibitors (cOmplete ULTRA tablets Mini EDTA free, Roche) and lysed by incubation with rotation for 10 min at room temperature. The lysate was cleared from cell debris by centrifugation (16,000 × g, 20 min, 4 °C). The cleared lysate was removed and a 17 μl input sample was taken.

For the IP, the Immunoprecipitation Kit Dynabeads™ Protein G (Invitrogen) was used. Briefly, 110 μl Dynabeads Protein G (20 μl/

sample + 10%) were prepared by washing with 200 μl binding & washing buffer. Subsequently, the beads were resuspended in 176 μl binding & washing buffer and 10 μl of α-FLAG M2 antibody (F1804-1MG, Sigma) was added. After incubation with rotation for 10 min at room temperature, the supernatant was removed and the beads washed once with binding & washing buffer. Finally, the beads were resuspended in 390 μl of lysis buffer and 70 μl of bead and lysis buffer mix were transferred to 5 tubes and kept at 4 °C until use.

For IP, the supernatant was removed from the prepared beads and the lysate was added. After 2.5 h of incubation with rotation at 4 °C, the supernatant was removed and the unbound sample was taken. The beads were then washed 3× with washing buffer (4× bead volume, 80 μl). To elute, the beads were resuspended in 25 μl SDS sample buffer, and boiled for 10 min.

### Protein purification

**Twin-Strep-tag purification.** Twin-Strep-tag purifications were conducted using Strep-TactinXT high capacity according to manufacturer's instructions (iba lifesciences). Briefly, cells from 100 mL exponential phase cultures resuspended in 1.5 mL lysis buffer (buffer W with added cOmplete Mini protease inhibitor cocktail, Roche) were lysed by repeated sonication (10 × 20 s, 1 s + 1 s break, strength: 30%) on ice. In a cold room, cleared samples were loaded onto columns containing 300 μl Strep-TactinXT high capacity matrix and incubated for 30 min to enhance binding. The columns were washed 8 times with 300 μL buffer W each and samples were eluted in 10 fractions with 150 μL BXT (wash buffer W containing 50 mM biotin) each. Elution buffer for fractions 5 and 6 was incubated on the column for 30 min to increase yield. Elution fractions were frozen and kept at −20 °C.

**His-SciP purification.** Purification of His-SciP was done according to published procedures[50] with the following changes: BL21(DE3)/pLysS cells were used and transformed with pHis-SciP by heat shock. All culture media were supplemented with ampicillin. The pre-culture was inoculated with multiple colonies. After resuspension of cells in lysis buffer supplemented with phenylmethylsulfonyl fluoride (PMSF), lysozyme and benzonase, the suspension was kept on ice for 20 min. Cells were lysed by repeated sonication (8 × 20 s, 1 s + 1 s break, strength: 50%). Lysate was cleared by centrifugation at 43,550 × g and afterwards bound to 1 mL 50% slurry of Talon Metal Affinity Resin (TaKaRa) equilibrated in lysis buffer without PMSF, lysozyme and benzonase. The final protein sample was concentrated (Pall Advance Centrifugal Device, MWCO 3 kDa), adjusted to 100 μM and stored in aliquots at −80 °C. Whenever buffers contained PMSF, lysozyme, benzonase or DTT, they were added immediately from stock before use.

**Purification of LarA, LarA variants and FliX.** Purification of substrates was adapted from Holmberg et al.[54]. BL21-SI/pCodonPlus cells were transformed using pMF65-c5 (LarA), pMF58 (FliX), pMF89-c2 (LarA$^{Δ5}$), pMF88-c4 (LarA$^{DD}$), pMF81-c2 (LarA$^{D}$), pMF82 (LarA$^{AA}$) by electroporation and selected on LBON agar plates supplemented with kanamycin (Kan) and chloramphenicol (Chlor). Pre-cultures (LBON or 2xYTON + Kan + Chlor) were inoculated with 20 colonies and cultivated at 30 °C overnight. 1 L main cultures in 2xYTON + Kan + Chlor were started by 1:100 dilution of the pre-culture und cultured until an OD approximately 1.0–1.5. Temperatures were then shifted to 20 °C and expression was started after 1 h by addition of 0.5 mM IPTG and 0.3 M NaCl (final concentrations). Incubation continued at 20 °C overnight (up to 20 h) and cells were harvested subsequently by centrifugation (6800 × g, 4 °C, 10 min) and cell pellets stored at −80 °C.

For purification, pellets were resuspended in HK500MG (40 mM HEPES-KOH pH7.4, 500 mM KCl, 5 mM MgCl, 5% glycerol), supplemented with 1 mM PMSF, 1 mg/mL Lysozyme and 3 μL Benzonase/10 mL and topped up to 20 mL total volume. Cells were then lysed by

2–4 passes through an EmulsiFlex-C3 high-pressure homogenizer and peak pressure was kept between 25,000 and 30,000 psi. Lysate was cleared by centrifugation at 32,500 × $g$ at 4 °C for 1 h. Tagged proteins were bound to 6 (=3 mL bed volume) or 3 mL (=1.5 mL bed volume) pre-equilibrated Talon SuperFlow Metal Affinity Resin from TaKaRa (LarA wt, LarA$^{AA}$, LarA$^{D}$ and LarA$^{DD}$, LarA$^{ΔS}$, respectively) or 1 g dry Protino Ni-IDA beads per liter culture (FliX) on ice for 30–90 min. After washing 5 times with approx. 45 mL HK500MG, bound proteins were eluted using HK500MG + 250 mM imidazole and fractions with protein concentrations ≥0.5 mg/mL were pooled. For 6×His-SUMO tag removal, 4 µg/mL Ulp1-6×His was added and imidazole was removed in parallel by dialysis against HK500G (40 mM HEPES-KOH pH7.4, 500 mM KCl, 5% glycerol). Tag depletion was achieved by binding to Talon Metal Affinity Resin (LarA wt, LarA$^{AA}$, LarA$^{D}$, LarA$^{DD}$, LarA$^{ΔS}$) or Protino Ni-IDA beads (FliX) as before and flow through containing purified protein was collected and a dialysed against storage buffer (10 mM HEPES-KOH pH8.0, 50 mM KCl, 0.1 mM EDTA, 1 mM DTT, 10% glycerol). Protein concentration was checked afterwards via SDS-PAGE (Bio-Rad 4–20% Mini-PROTEAN® TGX Stain-Free™ protein gel) and InstantBlue protein stain (Expedeon) or ReadyBlue (Sigma-Aldrich) and quantified using Bio-Rad ImageLab 6.0.1. In case of LarA$^{ΔS}$, tag depletion with Talon Metal Affinity Resin was insufficient. Therefore, an additional tag depletion using 0.5 g Protino Ni-IDA beads was performed and analysed as before. When necessary, proteins were concentrated using a Pall Advanced Centrifugal Device or an Amicon Ultra Filter with a MWCO 3 kDa (around 1/3rd of protein). Protein concentrations >100 µM were diluted to 100 µM, aliquoted, snap-frozen in liquid nitrogen and stored at −80 °C.

**Lon-His purification.** *E. coli* ER2566 was transformed using pBAD33-Lon6his and selected on LB Agar + Chlor at 37 °C. Pre-cultures (LB + Chlor) were inoculated with multiple colonies and cultured at 37 °C overnight. Main cultures of 1 L LB + Chlor were inoculated 1:100 grown at 37 °C. At an OD$_{600}$ of ~1.0 Lon-His expression was induced by 0.2% (w/v) L-arabinose for 3 h. Subsequently, cells were harvested by centrifugation (6800 × $g$, 4 °C, 10 min) and stored at −80 °C.

Pellets were resuspended in 60 mL lysis buffer (40 mM HEPES-KOH pH7.5, 250 mM NaCl, 10% glycerol), followed by addition of benzonase (3 µL per 10 mL) and lysozyme (1 mg/mL), cells lysed by passing through an EmulsiFlex-C3 high-pressure homogenizer twice and the resulting lysate was cleared by centrifugation at 18,000 × $g$ at 4 °C for 30 min. Afterwards, the NaCl concentration of the cleared lysate was adjusted to 500 mM and the cleared lysate was applied to 10 mL (=5 mL bed volume) equilibrated Talon SuperFlow Metal Affinity Resin (TaKaRa) per L culture on ice for 45 min. Subsequently, beads were washed 5 times with washing buffer (40 mM HEPES-KOH pH7.5, 500 mM NaCl, 10% glycerol, 10 mM imidazole). Finally, bound proteins were eluted in 1.5 mL fractions with elution buffer (40 mM HEPES-KOH pH7.5, 10% glycerol, 500 mM NaCl, 250 mM imidazole). Protein concentration was determined using a NanoDrop and purity was checked via SDS-PAGE (as described above). Elution fractions with >0.1 mg/mL were pooled and dialyzed twice against 50 mM HEPES-KOH pH7.5, 1 mM EDTA, 20% glycerol until imidazole concentration reached a calculated value <1 mM. Subsequently, a final concentration of 1 mM DTT was added, the protein solutions were concentrated using a Pall Advance Centrifugal Device (MWCO 30 kDa) and aliquots were snap-frozen and stored at −80 °C.

**Purification of Lon$^{EQ}$.** Purification of Lon$^{EQ}$ was done by expressing His-SUMO-LonEQ from pMF79-c5 as done for LarA, LarA variants and FliX (see above) with the following changes:

Cells were cultured in 2× 1 L cultures. Lysis buffer HN500G (40 mM HEPES-KOH pH 7.5, 500 mM NaCl, 10% glycerol) contained 0.1 mM EDTA and 0.05 mM DTT. After centrifugation, a final concentration of 10 mM imidazole was added to the cleared lysate and

subsequently was bound to 20 mL (=10 mL bed volume) Talon Metal Affinity Resin that was equilibrated in HN500G without EDTA and DTT. Five washes were performed using HN500G without EDTA/DTT. Elution was done with HN500G + 0.1 mM EDTA + 0.05 mM DTT + 250 mM imidazole.

Afterwards, the His-SUMO tag was cleaved by adding a final concentration of 4 µg/ml Ulp1-6×His and the solution is dialyzed simultaneously against buffer A (50 mM Tris/Cl pH 8.0, 50 mM KCl, 10% glycerol) over night. The recovered protein solution was further purified via anion exchange chromatography.

**Anion exchange of Lon$^{EQ}$.** Ion exchange chromatography was performed on an Äkta system using a Resource Q 1 ml column at a flow rate of 1 ml/min. After binding proteins and washing column using buffer A (50 mM Tris/Cl pH 8.0, 50 mM KCl, 10% glycerol), buffer B (50 mM Tris/Cl pH 8.0, 1000 mM KCl, 10% glycerol). Contaminating proteins were eluted at 15% buffer B until UV absorption reached baseline again. Similarly, Lon was eluted with 19% buffer B afterwards. Fractions were analysed via SDS-PAGE and visualized via ReadyBlue Protein Gel Stain (Sigma-Aldrich). Fractions containing Lon were pooled dialyzed against Lon storage buffer (50 mM HEPES-KOH pH 7.5, 1 mM EDTA, 1 mM DTT, 20% glycerol) over night. The recovered solution was subsequently concentrated using a 30 kDa centrifugal filter, insoluble Lon was spun down, the quality and concentration of Lon in the supernatant was assessed using SDS-PAGE and ReadyBlue-based staining, aliquoted in 20 µL, snap frozen and stored at −80 °C.

**Purification of 6xHis-titinI27 variants.** Purification of the 6xHis-titinI27 variants was done similar to the LarA purification. BL21-SI/pCodonPlus cells were transformed with pSH21-6xHis-titinI27-β20 (*6xHis-titinI27-β20*), pAK002 (*6xHis-titinI27-larA5*), pAK003 (*6xHis-titinI27-larA10*), pAK004 (*6xHis-titinI27-larA20*), pAK005 (*6xHis-titinI27-larA5-V88D-H89D*) or pAK006 (*6xHis-titinI27-LarA5-V86A-V88A*), pAK007 (*6xHis-titinI27-larA5-H89D*), pAK008 (*6xHis-titinI27*) and cells were grown in media containing ampicillin and chloramphenicol. Inductions were performed at 30 °C at an OD$_{600}$ of 1.0 using 0.5 mM IPTG and 0.3 M NaCl final concentration. Cleared lysate was bound to 3 mL (=1.5 mL bed volume) Talon SuperFlow Metal Affinity Resin (TaKaRa) per protein preparation. For lysis and wash buffer HNG10Im (40 mM HEPES-KOH pH7.5, 500 mM NaCl, 10% glycerol, 10 mM imidazole) was used as well as buffer HNG250Im for elution. Pall Advance Centrifugal Devices (MWCO 1 kDa) were used for buffer exchange to HG (25 mM HEPES-KOH pH7.5, 10% glycerol) and concentration of protein solution.

**Carboxymethylation of 6xHis-titin-I27 variants**
The purified 6xHis-titin-I27 proteins (see above) were bound to 1 g Protino Ni-IDA beads equilibrated with 25 mM Tris-HCl pH8.0 and washed a few times using the equilibration buffer. Iodoacetate label was prepared using 100 mM Tris-HCl pH8.0, 10 mM iodoacetate and 6 M guanidine hydrochloride in 10 mL water. The protein-bound beads were incubated with 3× volume of iodoacetate label for 3 h at room temperature protected from light on a rocker. The beads were then transferred to an empty PD-10 column and washed using 25 mM Tris-HCl pH8.0 a few times to remove the residual label. An elution buffer of 25 mM Tris-HCl pH8.0 and 250 mM imidazole was used to elute the proteins. Buffer exchange was carried out to remove imidazole. The proteins were stored in 25 mM HEPES-KOH pH7.5 and 10% glycerol. The concentrations were determined using BSA standards on SDS-PAGE.

**LarA antibody production**
Antibodies were produced by Davids Biotechnologie GmbH using purified LarA conjugated to a carrier (KLH) as an antigen. Polyclonal antiserum was generated in one rabbit by 5 immunizations.

Subsequently, LarA extracted from a 7 cm prep well Mini-PROTEAN® TGX Stain-Free™ gel (#4568091, Bio-Rad) was used for affinity purification of the final antibody.

## In vitro degradation assay

In vitro degradation assays were performed as described previously[29]. The reaction was carried out in Lon reaction buffer (25 mM Tris-HCl pH8.0, 100 mM KCl, 10 mM MgCl$_2$, 1 mM DTT) employing 0.75 μM Lon-His (=0.125 μM Lon$_6$), 4 μM substrate (if not stated otherwise) and an ATP regeneration system (4 mM ATP, 15 mM creatine phosphate, 75 μg/mL creatine kinase). The reaction mix and the ATP regeneration system were prepared separately and pre-warmed to 30 °C (~4 min). The reaction was started by adding the ATP regeneration system to the reaction mix. Samples were taken at indicated time points and quenched by 1 vol. 2× SDS loading buffer or 2× Tricine loading buffer (200 mM Tris-HCl, pH6.8, 2% SDS, 40% glycerol, 0.04% Coomassie Brilliant Blue G-250, 2% β-mercaptoethanol) and snap frozen in liquid nitrogen. Samples were heated at 65 °C for 10 min and separated by SDS-PAGE (Bio-Rad 4–20% Mini-PROTEAN® TGX Stain-Free™ protein gel) or Tris-Tricine based SDS-PAGE (16% Mini-PROTEAN® Tris-Tricin gel), visualized by InstantBlue protein stain (Expedeon) or ReadyBlue (Sigma-Aldrich) and quantified using Bio-Rad ImageLab 6.0.1. Substrate levels were normalised to Lon levels.

## Analysis of in vitro degradation rates

All degradation rates were determined in RStudio using R (v4.2 – v4.3.1) and the function nls of the stats package. The quality of the fitted curve was assessed in two ways. First the experimental data and the fitted model was plotted and assessed visually. Second the function 'summary' was used on the fitted model and the significance (p-values) of the fitted parameters as well as the residual standard error was used for further assessment of the model. For Fig. 3c, the degradation rates of His-SciP were determined as a function of LarA concentration and analysed by fitting the experimental data to Eq. (1) which was derived from Eq. (7) of Walsh et al.[55]

$$r_{deg} = V_b + (V_{\max} - V_b)\frac{c_{LarA}^n}{K_a^n + c_{LarA}^n} - (V_{\max} - V_i)\frac{c_{LarA}^n}{K_i^n + c_{LarA}^n} \qquad (1)$$

r$_{deg}$ corresponds to the degradation rate and c$_{LarA}$ to the LarA concentration. The starting values for each parameter were as following: $V_b = 1.3$, $V_{\max} = 8$, $K_a = 0.25$, $n = 1.7$, $V_i = 7$ and $K_i = 5$.

For Fig. 4d, the degradation rates of His-SciP were determined as a function of His-SciP concentration and analysed by fitting to a standard Michaelis-Menten equation (2) or Hill equation (3):

$$r_{deg} = V_{\max} * \frac{c_{SciP}}{(K_m + c_{SciP})} \qquad (2)$$

$$r_{deg} = V_{\max} * \frac{c_{SciP}^n}{(K_m^n + c_{SciP}^n)} \qquad (3)$$

Here, r$_{deg}$ corresponds to the degradation rate and c$_{SciP}$ to the His-SciP concentration. The starting values for V$_{\max}$ and K$_m$ were set to a value equal to half of the maximum measured degradation rate and half of the maximum tested His-SciP concentration, respectively. The starting value of the Hill coefficient was $n = 1$.

## ATPase assay

Lon ATPase rates were indirectly determined using a modified version of a previously published protocol[56], in which rates are determined by measuring oxidation of NADH to NAD$^+$ by lactate dehydrogenase (LDH) that oxidizes pyruvate, which itself is a byproduct of the ATP regeneration system. The reactions were carried out in Lon reaction buffer (see in vitro degradation assays) including 10% glycerol. The

ATP regeneration system (2 mM ATP, 2 mM phosphoenolpyruvate [PEP, Sigma-Aldrich, #P7127], 0.4 mM NADH [Roche, 10107735001], 1.67% (v/v) pyruvate kinase/lactate dehydrogenase mix [PK/LDH, Sigma-Aldrich, #P0294]) and the protease-substrate mix (0.05 μM Lon-His hexamer, substrates as indicated) were prepared separately in black Greiner UV-Star 96-well plates (#655809) and pre-warmed at 30 °C for 10 min. Afterwards, the solutions were mixed and measured immediately in a Spark microplate reader (Tecan) as follows: After shaking briefly (5 s, double orbital, 1 mm amplitude), absorption at 900 nm and 1000 nm (10 flashes, 50 ms settle time) for path length correction was measured once, followed by 300 cycles of measurements of absorption at 340 nm (25 flashes, 500 ms settle time) and shaking (as before). The analysis of the raw measurements were carried out in RStudio (https://www.rstudio.com) using R version 4.2.0 (markdown notebook provided at github.com/Matthionine/DetermineATPaseRate). In brief, the raw values were corrected to a path length of 1 cm and afterwards the blank was subtracted. Afterwards, an approach using a rolling window of 40 (20 or 10 for fast reactions) sequential measurements in combination with linear regression analysis was used to determine the slope of the data, which corresponds to the turnover of NADH per time unit. Then, the range with a somewhat constant minimum slope was chosen manually and the average of that range was used to calculate the maximum rate of ATP hydrolysis using the following formula:

$$r_{ATPase} = \frac{\frac{1}{\varepsilon_{NADH} * c_{ATPase} * d} * d(A_{340})}{dt} \qquad (4)$$

Where ε$_{NADH}$ = 6.220 × 10$^3$ M$^{-1}$ cm$^{-1}$, c$_{ATPase}$ corresponds to the hexameric concentration of Lon and $d = 1$ cm. Finally, the rate of NADH oxidation of PK/LDH without Lon was subtracted from all samples to get the ATPase rates of Lon-His.

## Flow cytometry

Cells of *C. crescentus* cultures grown in the indicated conditions were fixed in 70% ethanol. Fixed cells were pelleted at 4000 rpm, resuspended in 50 mM sodium citrate buffer containing 2 μg/mL RNase and incubated at 50 °C for 4 h or overnight to digest RNA. Samples were diluted and stained with 2.5 μM SYTOX green before being analysed by flow cytometry using a BD LSRFortessa flow cytometer (BD Biosciences). For each sample 30,000 cells were analysed. The experiments were performed in biological replicates and representative results are shown. The obtained data were analysed and processed with FlowJo software. Flow cytometry profiles within one figure were recorded in the same experiment, on the same day with the same settings. The scales of y- and x-axes of the histograms within one figure panel are identical. Each experiment was repeated independently and representative results are shown.

## Native mass spectrometry

Purified LonEQ (10 μM), LarA (60 μM), LarAΔ5 (60 μM) and SciP (40 μM) were buffer exchanged into 200 mM ammonium acetate pH 8.0, using Zeba Spin desalting columns, 7k MWCO (Thermo Fisher Scientific, USA). Samples were mixed in equimolar ratios, if not stated differently, and introduced into the mass spectrometer using offline nano ESI capillaries for mass spectrometry (Thermo Fisher Scientific, USA). Mass spectra were acquired on a Waters Synapt G2 travelling wave ion mobility mass spectrometer modified for high-mass analysis (MS Vision, NL) equipped with an offline nanospray source. The capillary voltage was set to 1.5 kV, the sample cone voltage 150 V, the source pressure was 8.3 mbar, the ion trap voltage 50 V, and the source temperature was 30 °C. Mass spectra were visualised using MassLynx 4.1 (Waters, UK) and MaCSED (http://benesch.chem.ox.ac.uk/resources.html) was used for assigning the charge sates and mass errors to the spectra.

## Structure prediction of LarA

The structures of LarA and its variants were predicted using ColabFold: AlphaFold2 using MMseqs2 (version v1.2)[35,36] provided as a Google Colab notebook. The prediction was run using standard settings: msa_mode: MMseqs2 (UniRef+Environmental); model_type: auto; num_models: 5; num_recycles: 3; use_amber: off; use_templates: off. The resulting predictions were analysed in Open-Source PyMOL.

## Sequence alignment of LarA orthologues

The Alignment of amino acid residue sequences of LarA and its homologues from different species was created using EMBL-EBI Multiple Sequence alignment tool MUSCLE[57]. Amino acid sequences were retrieved from ncbi.nlm.nih.gov.

## Statistical analysis

Data are shown as arithmetic means, error bars represent standard deviations (SD) and the sample sizes are stated in the corresponding figure legends. GraphPad Prism 9.4.1 and Microsoft Excel 2016 were used for data visualization and analysis. Statistical analysis was done in R (v4.2.1) using the rstatix package (v0.7.0). Differences between degradation rates of samples with and without LarA (Fig. 6b) were analysed by Welch's two-tailed, unpaired $t$-test using the t_test function using a confidence interval of 0.95 and assuming normal distribution and unequal variance for each condition.

## Reporting summary

Further information on research design is available in the Nature Portfolio Reporting Summary linked to this article.

## Data availability

All relevant data supporting the findings of this study are provided in the main figures and Supplementary Information files, and are available from the corresponding author upon request. The mass spectrometry proteomics data have been deposited to the ProteomeXchange Consortium via the PRIDE[58] partner repository with the database identifier PXD036514. Source data underlying Figs. 1c–f, 2a–f, 3a–e, 4b–g, 5b, c, 6a–d, 7c–g and Supplementary Figs. 1b, c, 2b, 3a, b, 4a, 6a–c, 7a–c and 8a–i are provided as Source Data files. Supplementary Datasets 1 and 2 can be found in the Source Data zip folder. Source data are provided with this paper.

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

## Acknowledgements

We thank the current and past members of the Jonas group for fruitful discussions and especially Roya Akar for excellent technical assistance. We also thank the Clinical Proteomics Mass Spectrometry facility, Karolinska Institute/Karolinska University Hospital/SciLifeLab for support and advice as well as the Drug Discovery and Development Platform at SciLifeLab, especially Anders Olsson and Yasmin Andersson, for sharing their equipment and experience in protein biochemistry. Furthermore, we thank Claes Andréasson, Peter Chien, Samar Mahmoud, Axel Mogk, Panagiotis Katikaridis and Karen Schriever for kindly sharing materials, protocols and for their advice. The study was funded by project grants from the Swedish Research Council (K.J.: 2016-03300, 2020-03545; M.L.: 2019-01961), a future leaders grant from the Swedish Foundation for Strategic Research (SSF, FFL15-0005) as well as and funding from the Strategic Research Area (SFO) program distributed through Stockholm University.

## Author contributions

Conceptualization, D.J.O., M.J.F., K.J.; Methodology, D.J.O, M.J.F., A.K., A.L.; Formal analysis, D.J.O., M.J.F., A.K., A.L., M.X.Z., Investigation, D.J.O., M.J.F., A.K., A.L., M.X.Z.; Writing—Original Draft, D.J.O., K.J.; Writing— Review & Editing, D.J.O., M.J.F., A.K., A.L., M.L., K.J.; Visualisation, D.J.O., M.J.F., K.J., Supervision, D.J.O., M.L., K.J.; Project Administration, K.J., Funding Acquisition, K.J.

## Funding

## Competing interests

The authors declare no competing interests.
