## [Peer Review File · Nature Communications]

The heat shock protein LarA activates the Lon protease in response to proteotoxic stressEditorial Note: This manuscript has been previously reviewed at another journal that is not operating a transparent peer review scheme. The manuscript was considered suitable for publication without further review at Nature Communications.

Reviewers' Comments:

Reviewer #1:

Remarks to the Author:

The authors report on the identification and characterization of LarA, a novel regulator of the AAA+ protease Lon from *C. crescentus* Lon activity. LarA is shown to allosterically increase Lon ATPase activity and to enhance proteolytic activity in a substrate-specific manner. LarA targets itself also for Lon-dependent degradation, thus limiting its stimulatory effects to the presence of selected substrates. LarA expression is strongly increased upon various stress conditions, leading to its transient accumulation as Lon-dependent degradation counteracts sustained LarA accumulation. This underlines (i) that activation of Lon by LarA must be tightly controlled and (ii) Lon activation through LarA is restricted to immediate stress periods. A growth phenotype of *larA* knockouts have so far not been identified by the authors, which is explainable by the highly transient nature of LarA accumulation.

The study is well executed and controlled and includes novel and interesting findings. How exactly LarA controls Lon activity and modulates substrate specificity remains unclear. While this certainly represents a mechanistic shortcoming, the new knowledge gained through this study weighs heavily and justifies publication.

Comments:

Do the authors also observe a transient LarA accumulation under stress conditions different from heat shock? Do they identify any phenotypes of *larA* mutants under these alternative stress regimes?

Reviewer #2:

Remarks to the Author:

In this revision of a revised manuscript, the authors addressed a minor concern about the methods. However, the manuscript as written still requires some minor corrections. In particular, the claim that LarA activates Lon proteolysis at the onset of proteolytic stress has not been shown experimentally. In addition, the results showing that $\Delta larA$ have no viability defects should be shown in supplemental data and not as 'data not shown', because it is important to show this observation. If these minor issues are addressed, this revised manuscript would be a valuable contribution to the field.

Detailed comments:

In the abstract, the authors state that 'LarA is degraded by Lon itself, which is critical to prevent toxic overactivation of Lon.'. Demonstrating that degradation by Lon is important for preventing LarA overactivation of Lon was not shown in the manuscript as written, nor was it shown that LarA overexpression inhibits growth because of Lon overactivation, just that LarA overexpression has no effect in Δlon . It is suggested that this sentence be rewritten as something similar to 'Further, we find that LarA is degraded by Lon itself and overexpression of LarA results in toxicity dependent on the presence of Lon.'

On line 83 of the introduction, the authors state that LarA is 'a novel dedicated Lon regulator in *C. crescentus* that activates Lon proteolysis at the onset of proteotoxic stress.' However, the experiments

to show that LarA actually activates Lon proteolysis upon proteotoxic stress are not presented in the manuscript as written. It is suggested that the phrase 'at the onset of proteotoxic stress' be removed as it is clear that LarA can activate Lon proteolysis, it is just unknown whether it does so specifically at the onset of stress. This revised sentence would end in with: 'a novel dedicated Lon regulator in *C. crescentus* that activates Lon proteolysis.'

The results showing that Δ larA has no viability issues (lines 469-471) should be published as supplemental results, rather than have them stated as data not shown. The fact there are no obvious growth defects supports the authors' model that there are redundant mechanisms involved during this response.

Point-to-point response to reviewers (manuscript # # NCOMMS-23-36700A)

We thank the reviewers for their comments on our revised manuscript. We have addressed each point either by including additional data or by modifying the text. Please find our responses below, with the original points appearing in black and our replies in blue.

Reviewer #1:

The authors report on the identification and characterization of LarA, a novel regulator of the AAA+ protease Lon from *C. crescentus* Lon activity. LarA is shown to allosterically increase Lon ATPase activity and to enhance proteolytic activity in a substrate-specific manner. LarA targets itself also for Lon-dependent degradation, thus limiting its stimulatory effects to the presence of selected substrates. LarA expression is strongly increased upon various stress conditions, leading to its transient accumulation as Lon-dependent degradation counteracts sustained LarA accumulation. This underlines (i) that activation of Lon by LarA must be tightly controlled and (ii) Lon activation through LarA is restricted to immediate stress periods. A growth phenotype of *larA* knockouts have so far not been identified by the authors, which is explainable by the highly transient nature of LarA accumulation.

The study is well executed and controlled and includes novel and interesting findings. How exactly LarA controls Lon activity and modulates substrate specificity remains unclear. While this certainly represents a mechanistic shortcoming, the new knowledge gained through this study weighs heavily and justifies publication.

Comments:

Do the authors also observe a transient LarA accumulation under stress conditions different from heat shock? Do they identify any phenotypes of *larA* mutants under these alternative stress regimes?

Following this reviewer's suggestion, we have analysed LarA levels in response to EtOH, canavanine (Can) and azetidine-2-carboxylic acid (AzC) stress over an extended period of time to see if LarA accumulation is transient. Under EtOH stress, LarA levels peak shortly after stress exposure and subsequently decrease again, thus showing a pattern resembling the transient accumulation of LarA under heat stress. Interestingly, we observed that Can and AzC treatment caused an accumulation of LarA that persisted for a longer period of time.

We agree with the reviewer that the identification of stress conditions, in which LarA accumulation is not as transient, may help to detect growth phenotypes of a *larA* mutant. We have searched for a phenotype of the $\Delta larA$ mutant under a range of stress conditions, including during heat, EtOH, Can, AzC and kanamycin stress, but were so far not able to detect a clear growth phenotype. As explained in our previous point-to-point response and discussed in our revised manuscript, we think that this can likely be explained by redundant protein quality control mechanisms that contribute to protein homeostasis when Lon regulation by LarA is absent. It is also noteworthy, that absence of Lon itself results in rather mild growth defects, supporting the idea that other proteases and chaperones take over when one key component of the network is missing.

We have included the new Western blot data and the phenotypic data for WT, Δlon and $\Delta larA$ cells under the different stress conditions as a new Supplementary Figure 8 in the revised version and have modified the text accordingly. Furthermore, we also changed in several places, including the title, the text from "at the onset of proteotoxic stress" to "in response to proteotoxic stress".

Reviewer #2:

In this revision of a revised manuscript, the authors addressed a minor concern about the methods. However, the manuscript as written still requires some minor corrections. In particular, the claim that LarA activates Lon proteolysis at the onset of proteolytic stress has not been shown experimentally. In addition, the results showing that $\Delta larA$ have no viability defects should be shown in supplemental data and not as 'data not shown', because it is important to show this observation. If these minor issues are addressed, this revised manuscript would be a valuable contribution to the field.

Detailed comments:

In the abstract, the authors state that 'LarA is degraded by Lon itself, which is critical to prevent toxic overactivation of Lon.'. Demonstrating that degradation by Lon is important for preventing LarA

overactivation of Lon was not shown in the manuscript as written, nor was it shown that LarA overexpression inhibits growth because of Lon overactivation, just that LarA overexpression has no effect in Δ lon. It is suggested that this sentence be rewritten as something similar to 'Further, we find that LarA is degraded by Lon itself and overexpression of LarA results in toxicity dependent on the presence of Lon.'

As the reviewer suggested, we have modified the statement about LarA degradation by Lon and LarA's toxic effects. The sentence reads now "Further, we find that high levels of LarA cause growth inhibition in a Lon-dependent manner and that Lon-mediated degradation of LarA itself ensures low LarA levels in the absence of stress."

On line 83 of the introduction, the authors state that LarA is 'a novel dedicated Lon regulator in *C. crescentus* that activates Lon proteolysis at the onset of proteotoxic stress.' However, the experiments to show that LarA actually activates Lon proteolysis upon proteotoxic stress are not presented in the manuscript as written. It is suggested that the phrase 'at the onset of proteotoxic stress' be removed as it is clear that LarA can activate Lon proteolysis, it is just unknown whether it does so specifically at the onset of stress. This revised sentence would end in with: 'a novel dedicated Lon regulator in *C. crescentus* that activates Lon proteolysis.'

We have changed this sentence based on the reviewer's suggestion to "Here, we report the discovery and characterization of LarA (Lon activity regulator A), a dedicated and stress-induced Lon regulator in *C. crescentus* that activates Lon proteolysis of a broad range of substrates."

The results showing that Δ larA has no viability issues (lines 469-471) should be published as supplemental results, rather than have them stated as data not shown. The fact there are no obvious growth defects supports the authors' model that there are redundant mechanisms involved during this response.

We agree with the reviewer. We have added the phenotypic data in panels b)-i) of our new Supplementary Figure 8, and describe them in the last section of the results.